# Interannual variation, decadal trend, and future change in ozone outflow from East Asia

Jia Zhu[1,2] , Hong Liao[3,4] , Yuhao Mao[3], Yang Yang[5], and Hui Jiang[6]

[1]State Key Laboratory of Atmospheric Boundary Layer Physics and Atmospheric Chemistry (LAPC), Institute of Atmospheric Physics, Chinese Academy of Sciences, Beijing, China

[2]University of Chinese Academy of Sciences, Beijing, China

[3]School of Environmental Science and Engineering, Nanjing University of Information Science & Technology, Nanjing, China

[4]Joint International Research Laboratory of Climate and Environment Change, Nanjing University of Information Science & Technology, Nanjing, China

[5]Atmospheric Science and Global Change Division, Pacific Northwest National Laboratory, Richland, Washington, USA

[6]National Meteorological Information Center, China Meteorological Administration, Beijing, China

*Correspondence to:* Hong Liao (hongliao@nuist.edu.cn)

**Abstract.** We examine the past and future changes in $O_3$ outflow from East Asia using a global three-dimensional chemical transport model GEOS-Chem. The simulations of Asian $O_3$ outflow for 1986–2006 are driven by the assimilated GEOS-4 meteorological fields, and those for 2000–2050 are driven by the meteorological fields archived from the Goddard Institute for Space Studies (GISS) General Circulation Model (GCM) 3 under the IPCC SRES A1B scenario. Evaluation of the model results against measurements shows that the GEOS-Chem model captures fairly well the seasonal cycles and interannual variations of tropospheric $O_3$ concentrations, with high correlation coefficients of 0.82–0.93 at four ground-based sites and of 0.55–0.88 at two ozonesonde sites where observations are available. The increasing trends in surface-layer $O_3$ concentrations in East Asia over the past two decades are captured by the model, although the modeled $O_3$ trends have low biases. Sensitivity studies are conducted to examine the respective impacts of meteorological parameters and emissions on the variations in the outflow flux of $O_3$. When both meteorological parameters and anthropogenic emissions varied during 1986–2006, the simulated Asian $O_3$ outflow fluxes exhibited a statistically insignificant decadal trend, but large interannual variations (IAVs) with seasonal absolute percent departure from the mean (APDM) values of 4–9 % and annual APDM value of 3.3 %. Sensitivity simulations indicated that the large IAVs of $O_3$ outflow fluxes were mainly caused by the variations in meteorological conditions. Variations in meteorological parameters drove the IAVs in $O_3$ outflow fluxes by altering $O_3$ concentrations over East Asia and by altering zonal winds, and the latter was identified to be the key factor since the $O_3$ outflow was highly correlated with zonal winds during 1986–2006. The simulations of the 2000–2050 changes show that the annual outflow flux of $O_3$

will increase by 2.0 %, 7.9 %, and 12.2 %, respectively, owing to climate change alone, emissions change alone, and changes in both climate and emissions. Therefore, climate change will aggravate the effects of the increases in anthropogenic emissions on future changes in the Asian $O_3$ outflow. Future climate change is predicted to greatly increase Asian $O_3$ outflow in the spring and summer seasons as a result of the projected increases in zonal winds.

Findings from the present study help to understand the variations in tropospheric $O_3$ in the downwind regions of East Asia on different timescales, and have important implications for long-term air quality planning for the downwind regions of China, such as Japan and US.

## 1 Introduction

Tropospheric ozone ($O_3$) is an important air pollutant, which has a detrimental effect on human health (Fann et al., 2012; Jhun et al., 2014), crops (Wilkinson et al., 2011; Tai et al., 2014), and ecosystems (Fuentes et al., 2013; Yue and Unger, 2014). It is also an important greenhouse gas that directly contributes to global warming (IPCC, 2013). $O_3$ has a relatively long lifetime of weeks in the free troposphere (Young et al., 2013; Monks et al., 2015), which makes intercontinental transport of $O_3$ an important issue for understanding $O_3$ concentrations and planning emission control

measures.

    A number of previous studies have shown that Asian continental outflow impacts the global $O_3$ budget (Liu et al., 2002), and influences $O_3$ air quality in the downwind regions, such as the western North Pacific through the western North America (Jacob et al., 1999; Tanimoto et al., 2005; Kim et al., 2006; Li et al., 2008; Zhang et al., 2008; Chiang et al., 2009; Kurokawa et al., 2009; Huang et al., 2010; Nagashima et al., 2010; Walker et al., 2010; Ambrose et al.,

2011; Lin et al., 2012; Ou-Yang et al., 2013; Han et al., 2015; Pochanart et al., 2015). Liu et al. (2002) reported that boundary-layer $O_3$ pollution was lifted into the upper troposphere by deep convection over the Asian maritime continent, from where it was transported northward along the upper branch of the local Hadley circulation and into the mid-latitude westerlies, influencing the global $O_3$ budget. Using a global 3-D chemical transport model GEOS-Chem, Zhang et al. (2008) estimated that Asian pollution enhanced surface-layer $O_3$ concentrations by 5–7 ppbv over western

North America in spring 2006. Walker et al. (2010) used the same model (GEOS-Chem) to evaluate sensitivities of tropospheric $O_3$ over Canada to Asian anthropogenic emissions, and reported that the contribution from Asian emissions to $O_3$ profiles above Whistler, Canada, was 6–8 ppbv in spring 2006. Through an integrated analysis of in situ and satellite measurements in May–June 2010 with a global chemistry-climate model GFDL AM3, Lin et al. (2012) reported that Asian emissions could contribute 8–15 ppbv $O_3$ over the western United States on days when the

observed daily maximum 8-h average $O_3$ (MDA8 $O_3$) exceeded 60 ppbv, and that 20 % of MDA8 $O_3$ exceedances of 60 ppbv would not have occurred in the southwestern United States in the absence of Asian anthropogenic emissions.

Asian $O_3$ outflow exhibits seasonal variations (Liu et al., 2002; Han et al., 2015). Using a global 3-D chemical transport model GEOS-Chem, Liu et al. (2002) simulated the seasonal variations of the Asian outflow flux of $O_3$ over the Pacific, which was defined as the eastward flux integrated for the tropospheric column through a wall located at $150\,°E$ between $10\,°N$ and $60\,°N$. They found that the Asian $O_3$ outflow flux reached the maximum in early spring (March) and the minimum in summer (July). Han et al. (2015) examined $O_3$ measurements at Ieodo Ocean Research Station, which was located in the East China Sea and regarded as an ideal place to observe Asian outflow without local effects. They reported that the seasonal variation of $O_3$ was distinct, with a minimum in August and two peaks in April and October, and was greatly affected by the seasonal wind pattern over East Asia.

Continental outflow of $O_3$ is expected to vary on interannual to decadal timescales, because tropospheric $O_3$ concentrations and meteorological parameters have variations on these timescales. Large interannual variations (IAVs) of tropospheric $O_3$ concentrations have been reported in previous observational studies (Kurokawa et al. 2009; Zhou et al., 2013). Analyzing 11 years of ozonesonde data over Hong Kong, Zhou et al. (2013) reported that observed tropospheric $O_3$ levels during 2000–2010 exhibited high IAV, with an annual averaged amplitude [defined as (maximum + 2nd maximum − minimum − 2nd minimum) $\times$ 0.5 / the average during 2000–2010] up to 30 % of the averaged concentrations at 3–8 km altitude. Kurokawa et al. (2009) analyzed observed springtime $O_3$ over Japan during 1985–2005, and found that the observed $O_3$ showed greater year-to-year variations than the annual rate of the long-term trend. Decadal trends of tropospheric $O_3$ concentrations have been reported for different locations on the basis of observations (Ding et al., 2008a; Xu et al., 2008; Tang et al., 2009; Tanimoto, 2009; Wang et al., 2009b; Cooper et al., 2010; Wang et al., 2012; Lin et al., 2014, 2015; Zhang et al., 2014), such as $-0.56$ ppbv $yr^{-1}$ over Linan in eastern China (Xu et al., 2008), $+0.58$ ppbv $yr^{-1}$ over Hong Kong in southern China (Wang et al., 2009b), $+1.0$ ppbv $yr^{-1}$ at Mt. Happo in Japan (for springtime $O_3$; Tanimoto, 2009), and $+0.35$ ppbv $yr^{-1}$ over Hawaii in North Pacific (for autumn $O_3$; Lin et al., 2014). Asian $NO_x$ emissions almost doubled over the past 20 years (Yang et al., 2015), which contributed to the raised $O_3$ observed over the downwind regions of Asia (Lin et al., 2016).

Future changes of tropospheric $O_3$ concentrations have also been predicted by modeling studies (Racherla and Adams, 2006, 2009; Lin et al., 2008; Wu et al., 2008a; Lam et al., 2011; Wild et al., 2012; Gao et al., 2013; Liu et al., 2013; Wang et al., 2013; Lee et al., 2015; Val Martin et al., 2015; Schnell et al., 2016; Zhu and Liao, 2016). Wang et al. (2013), using the NASA GISS GCM/GEOS-Chem model combination, reported that the summer surface-layer $O_3$ levels averaged over China would increase by 11.9 ppbv due to the combined changes in climate and emissions over 2000–2050 under the SRES A1B scenario.

Meteorological parameters, especially winds that are important for $O_3$ outflow, also exhibit variations on different time scales (Chang et al., 2000; Ding et al., 2008b; Sun et al., 2009; Zhang and Guo, 2010; Hirahara et al., 2012).

Large IAVs of the East Asian summer monsoon (EASM) have been reported in previous studies (Zhu et al., 2012; Yang et al., 2014). The decadal-scale weakening of the EASM since the 1950s has also been reported by many previous studies, and anomalous northeasterlies during the weak monsoon years were found over the western North Pacific near 40 °N, which did not favor the outflow of pollutants from northern China (Chang et al., 2000; Ding et al., 2008b; Zhu et al., 2012). On the basis of NCEP/NCAR reanalysis data, Sun et al. (2009) showed that the axis location of the East Asia subtropical westerly jet (EASWJ) had displaced southward since the end of the 1970s, intensifying the westerly wind over 25–35 °N and weakening it over 42–50 °N, and therefore influencing the outflow of pollutants. Lin et al. (2014) reported that interannual variability in springtime Asian $O_3$ transport, as inferred by the East Asian COt (carbon-monoxide-like tracer), was strongly influenced by ENSO-related shifts in the subtropical jet stream, and that the decrease in ozone-rich Eurasian airflow reaching the eastern North Pacific during spring in the 2000s was attributed to more frequent La Nina events. Most of the Coupled Model Intercomparison Project Phase 3 (CMIP3) models projected that the Asian jet would be intensified on its equatorward side by the end of the 21st century (Zhang and Guo, 2010; Hirahara et al., 2012).

Few previous studies have examined the IAVs, decadal trends, and future changes in $O_3$ outflow. In this work, we examine the historical (1986–2006) and future (2000–2050) changes of $O_3$ outflow from East Asia, and systematically quantify the roles of meteorological parameters and/or anthropogenic emissions on the changes. The descriptions of the model, emissions, and numerical simulations are presented in Sect. 2. Section 3 evaluates the model performance for tropospheric $O_3$. Section 4 discusses the IAVs and decadal trends in the $O_3$ outflow from East Asia over 1986–2006. Future changes in $O_3$ outflow from East Asia for 2000–2050 are presented in Sect. 5.

## 2 Methods

### 2.1 Model description

We apply the global 3-D chemical transport model GEOS-Chem to simulate $O_3$ outflow fluxes. The GEOS-Chem model includes a detailed simulation of $O_3$–NOx–hydrocarbon (~80 species, ~300 chemical reactions) (Bey et al., 2001) and aerosol chemistry. Aerosol species include sulfate ($SO_4^{2-}$), nitrate ($NO_3^-$), ammonium ($NH_4^+$) (Park et al., 2004; Pye et al., 2009), black carbon (BC) and organic carbon (OC) (Park et al., 2003), mineral dust (Fairlie et al., 2007), and sea salt (Alexander et al., 2005). The simulations account for the impacts of aerosols on the distributions and concentrations of $O_3$ through heterogeneous reactions and changes in photolysis rates (Lou et al., 2014).

To simulate historical changes in the Asian outflow of $O_3$, the GEOS-Chem model is driven by the assimilated GEOS-4 meteorological fields from the Goddard Earth Observing System (GEOS) of NASA Global Modeling and

Assimilation Office (GMAO). We perform simulations for 1986–2006, which are the years with available GEOS-4 meteorological datasets. The version of the model used here has a horizontal resolution of 2° (latitude) × 2.5° (longitude), with 30 vertical layers.

To simulate future changes of $O_3$ outflow fluxes during 2000–2050, the GEOS-Chem simulation is driven by meteorological data from the National Aeronautics and Space Administration/Goddard Institute for Space Studies (NASA/GISS) general circulation model (GCM) 3 (Rind et al., 2007) for both the present day (1996–2005) and future (2046–2055), following Wu et al. (2008b), Pye et al. (2009), Wang et al. (2013), and Jiang et al. (2013). Both the GISS and GEOS-Chem models used here have a horizontal resolution of 4° latitude by 5° longitude, with 23 vertical layers.

## 2.2 Emissions

For simulations during 1986–2006, the global anthropogenic emissions of reactive nitrogen oxides ($NO_x$), carbon monoxide (CO), and sulfur dioxide ($SO_2$) in the model are from the Emission Database for Global Atmospheric Research (EDGAR) inventory (Olivier and Berdowski, 2001). The global emissions of non-methane volatile organic compounds (NMVOCs) are from the Global Emissions Inventory Activity (GEIA) inventory (Piccot et al., 1992). Global emissions of carbonaceous aerosols (BC and OC) follow Bond et al. (2007). Anthropogenic emissions of reactive $NO_x$, CO, $SO_2$, $NH_3$, and NMVOCs over East Asia are overwritten by the emissions inventory of Streets et al. (2003) and Zhang et al. (2009). IAVs of anthropogenic emissions are represented by global-gridded annual scaling factors as described by van Donkelaar et al. (2008) for $NO_x$, CO, and NMVOCs. Biomass burning emissions are taken from the Global Fire Emissions Database-3 (GFED-3) inventory (van der Werf et al., 2010) for 1997–2006. The biomass burning emissions before 1997 are unavailable because of the lack of datasets.

Figure 1 shows the evolution of anthropogenic and biomass burning emissions of $O_3$ precursors ($NO_x$, CO, NMVOCs) summed over the globe and Asia (60–150° E, 10° S–55° N) over 1986–2006. Global anthropogenic emissions of these precursors exhibited no significant trends, while the Asian anthropogenic emissions showed large increases over the past two decades. Relative to year 1986, the Asian anthropogenic emissions of $NO_x$, CO, and NMVOCs in 2006 increased by 70.0 %, 42.1 %, and 50.9 %, respectively. Compared with anthropogenic emissions, biomass burning emissions had greater IAVs during 1997–2006. Figure 1 also shows the pathway for the global $CH_4$ abundance used in our simulations of $O_3$. The $CH_4$ mixing ratio in 1986 was 1672 ppb, which increased by 6.3 % in 2006. Note that, over 1996–2006 when $NO_x$ emissions and satellite $NO_2$ columns were simultaneously available, the trend in $NO_x$ emissions over East-Central China (ECC, 110–123° E, 30–40° N) was +8.2% yr$^{-1}$ on the basis of the emission inventory used in this study, close to the trend of +9.0% yr$^{-1}$ in $NO_2$ columns averaged over ECC on the basis

of tropospheric $NO_2$ vertical column density (VCD) data retrieved from GOME (1996–2002) and SCIAMACHY (2003–2006), which are available from www.temis.nl.

For future simulations during 2000–2050, anthropogenic emissions of $O_3$ precursors, including $NO_x$, CO, and NMVOCs, are taken from Wu et al. (2008b), and those of $NH_3$ and $SO_2$ follow those in Pye et al. (2009). The future anthropogenic emissions of $O_3$ precursors, aerosol precursors, and aerosols under the SRES A1B scenario are generated by the Integrated Model to Assess the Greenhouse Effect (IMAGE) socioeconomic model using growth factors for different species and countries (Streets et al., 2004). Table 1 shows the present-day (year 2000) and future (year 2050) anthropogenic emissions of $O_3$ precursors under the SRES A1B scenario. The global emissions of $NO_x$, CO, and NMVOCs are projected to increase by 78.4 %, 26.4 %, and 89.4 % over 2000–2050; and the Asian emissions are projected to increase by 159.6 %, 23.7 %, and 118.6 %, respectively. Present-day methane mixing ratios are specified as 1750 ppbv on the basis of observations (Wu et al., 2008b). The future methane concentrations are set to 2400 ppbv, following the SRES A1B scenario (Pye et al., 2009).

The natural emissions of $O_3$ precursors, including $NO_x$ from lighting and soil, and NMVOCs from vegetation, are calculated on the basis of the assimilated GEOS-4 meteorological fields and GISS Model 3 meteorological parameters. The lightning $NO_x$ emissions follow Price and Rind (1992), with the $NO_x$ vertical profile proposed by Pickering et al. (1998). The soil $NO_x$ emissions scheme in the GEOS-Chem model is based on the work of Yienger and Levy (1995) and Wang et al. (1998). Biogenic emissions of NMVOCs are calculated according to the Model of Emissions of Gases and Aerosols from Nature (MEGAN) (Guenther et al., 2006). Figure 2 shows the evolution of natural emissions summed over the globe and Asia over 1986–2006. Both global and Asian $NO_x$ emissions from lightning exhibited large IAVs and significant increasing trends. It has been shown that warming climate leads to increased lightning $NO_x$ (IPCC, 2013). Compared with lightning $NO_x$ emissions, $NO_x$ emissions from soil showed smaller IAVs and no significant decadal trend. Both global and Asian biogenic emissions of NMVOCs have been shown to have large IAVs, as a result of the changes in both vegetation and meteorological parameters (Fu and Liao, 2012). For future simulations during 2000–2050, the natural emissions of $O_3$ precursors are listed in Table 2. The simulated emissions of lightning $NO_x$, soil $NO_x$, and biogenic VOCs are projected to increase by 18.8 %, 14.9 %, and 22.1 % for the globe, and by 16.7 %, 21.4 %, and 18.9 % for Asia, respectively.

The effects of changes in stratosphere–troposphere exchange (STE) of $O_3$ are not included in this study for both past and future simulations. The cross-tropopause $O_3$ flux is represented by the synthetic $O_3$ (Synoz) method (McLinden et al., 2000), which imposes a global annual mean cross-tropopause $O_3$ flux of 500 Tg yr$^{-1}$.

**2.3 Numerical experiments**

To examine the respective and combined impacts of meteorological parameters, anthropogenic emissions, and biomass burning emissions on the IAVs and decadal trends of $O_3$ outflow from East Asia during 1986–2006, we perform simulations for four cases (Table 3):

(a) Met: The simulation of $O_3$ outflow for 1986–2006, to examine the effect of variations in meteorological parameters alone. The meteorological parameters vary from 1986 to 2006, and the anthropogenic emissions are fixed at year 2006 levels. Biomass burning emissions are turned off.

(b) Emis: The simulation of $O_3$ outflow for 1986–2006, to examine the effect of variations in anthropogenic emissions alone. The anthropogenic emissions vary from 1986 to 2006, and the meteorological parameters are fixed at year 2006 levels. Biomass burning emissions are turned off.

(c) MetEmis: The simulation of $O_3$ outflow for 1986–2006, with variations in both meteorological parameters and anthropogenic emissions during 1986–2006. Biomass burning emissions are turned off.

(d) MetEmisB: The simulation of $O_3$ outflow for 1997–2006, with variations in meteorological parameters, anthropogenic emissions, and biomass burning emissions during 1997–2006. Note that biomass burning emissions in the model are not available before 1997.

To identify the relative roles of future changes in meteorological parameters and emissions in 2000–2050 changes in Asian $O_3$ outflow flux, another four simulations are carried out: (a) Met2000Emis2000: present-day climate and emissions; (b) Met2050Emis2000: future climate and present-day anthropogenic emissions; (c) Met2000Emis2050: present-day climate and future anthropogenic emissions; and (d) Met2050Emis2050: future climate and emissions. Both the future climate and anthropogenic emissions follow the IPCC SRES A1B scenario.

The mass flux of $O_3$ through the vertical plane along 135 °E from 20 °N to 55 °N from the surface to 100 hPa is used to quantify Asian $O_3$ outflow. The metric of mass flux through a vertical plane was also used by Liu et al. (2002) to represent Asian $O_3$ outflow, and by Jiang et al. (2013) and Yang et al. (2015) to represent Asian aerosol outflow. It should be noted that the $O_3$ outflow flux from East Asia includes the effects of emissions in different regions of the world owing to the relatively long lifetime (~3 weeks) of $O_3$ (Fiore et al., 2002; Liao et al., 2006). However, Liu et al. (2002) found that anthropogenic sources in Asia made the largest contribution to the Asian outflow flux of $O_3$.

**3 Model evaluation**

The GEOS-Chem simulations of $O_3$ have been evaluated extensively for the U.S. (Liu et al., 2006; Wu et al., 2008b; Zhang et al., 2008), Europe (Auvray and Bey, 2005; Liu et al., 2006; Kim et al., 2015), and China (Wang et al., 2013; Lou et al., 2014; Yang et al., 2014; Zhu and Liao, 2016). These studies showed that the GEOS-Chem model captured

the magnitude and distribution of the surface-layer concentration and column burden of tropospheric $O_3$ fairly well. The vertical distributions of $O_3$ have also been evaluated by aircraft campaigns and ozonesonde measurements (Zhang et al., 2008; Walker et al., 2010; Wang et al., 2011), showing that the GEOS-Chem model closely reproduced the observed $O_3$ profiles.

Here, we conduct comparisons with measurements to evaluate whether the version of the GEOS-Chem model used in this study can capture the temporal variations of tropospheric $O_3$. We use observations of tropospheric $O_3$ available in East Asia as summarized in Table 4. Observations at two sites (Minamitorishima and Yonagunijima) are from the World Data Centre for Greenhouse Gases (WDCGG, www.ds.data.jma.go.jp/gmd/wdcgg/), and those at another two sites (Rishiri and Ogasawara) are from the Acid Deposition Monitoring Network in East Asia (EANET, www.eanet.asia/product/index.html), which are used to evaluate the simulated surface-layer $O_3$ concentrations. The four Japanese sites are "remote" sites in the downwind regions of China. Figure 3 compares the time series of monthly surface-layer $O_3$ mixing ratios simulated by MetEmisB with those measured by WDCGG and EANET. Simulated surface-layer $O_3$ levels agree well with observations at all the four stations. The model captures fairly well the seasonal cycles and interannual variations of surface $O_3$, with high correlation coefficients of 0.82–0.93 (Table 4). Generally, the GEOS-Chem model can capture the high values during early spring or winter when Asian $O_3$ outflow flux is the highest, but overestimates the low values during summer when Asian $O_3$ outflow is the minimum.

To evaluate the simulated $O_3$ concentrations for the boundary layer, middle and upper troposphere, we use the ozonesonde data at two Japanese sites from World Ozone and Ultraviolet Radiation Data Centre (WOUDC, www.woudc.org). The information for the two sites (Naha and Tsukuba) is listed in Table 4. Figure 4 compares the time series of monthly $O_3$ mixing ratios simulated by MetEmisB with those measured by ozonesonde. Comparisons are shown for four altitudes in the troposphere. The GEOS-Chem model captures the seasonal cycles and interannual variations of tropospheric $O_3$ at all altitudes, with correlation coefficients ranging from 0.68 to 0.88 for Naha site, and from 0.55 to 0.76 for Tsukuba site. However, the agreement with ozonesonde in the lowermost layer (1000–850 hPa) seems to be poorer than that with WDCGG or EANET. It is noted that, the ground-based measurements (WDCGG or EANET) and simulation results are calculated from continuous data, while the ozonesondes are regularly launched at a fixed local time with a typical frequency of 1–2 weeks (Tanimoto et al., 2015). The inconsistency in sampling time may be responsible for the poorer agreement with ozonesonde.

The increasing trend in surface-layer $O_3$ in East Asia over the past two decades was reported by previous studies (Ding et al., 2008a; Wang et al., 2009b; Xu et al., 2016). Figure 5 compares the simulated trends in seasonal or annual mean surface-layer $O_3$ concentrations from the MetEmis experiment with the observed trends collected from previous studies. Simulated $O_3$ concentrations exhibit statistically significant increasing trends at all sites except for Waliguan in

winter, although the model underestimates the trends for some stations and seasons. The modeled $O_3$ trends were also reported to have low biases in previous studies (Tanimoto et al., 2009; Parrish et al., 2014; Strode et al., 2015). Parrish et al. (2014) compared $O_3$ trends simulated by three chemistry-climate models with observations at Asian sites, and reported that one model captured less than one third of the observed increasing trend whereas the other two models suggested no significant increasing trends.

In general, the GEOS-Chem model can capture fairly well the seasonal cycles and interannual variations of tropospheric $O_3$, although the model overestimates the low values during summer indicating an overestimation of Asian $O_3$ outflow in summer. The increasing trends in surface-layer $O_3$ in China over the past two decades can also be captured by GEOS-Chem model, although the modeled $O_3$ trends have low biases.

## 4 Simulated Asian $O_3$ outflow during 1986–2006

### 4.1 Seasonal patterns of Asian $O_3$ outflow

Figure 6 shows the pressure–latitude cross-sections along $135\,°E$ of the seasonal $O_3$ outflow fluxes averaged over 1997–2006 in the MetEmisB simulation. The maximum $O_3$ fluxes were found in the middle-upper troposphere, in consistent with Liu et al. (2002) and Wang et al. (2009a), and it could be attributed to the vertical distributions of both zonal winds and $O_3$ concentrations. The westerlies strengthen with altitudes with the strongest winds occurring around 200 hPa (known as the East Asia subtropical westerly jet) (Ren et al., 2011). Concentrations of $O_3$ are high in the upper troposphere over the mid-latitudes (Wang, 1999).

The seasonal mass fluxes through the meridional plane (along $135\,°E$ from $20\,°N$ to $55\,°N$, and from the surface to 100 hPa) were calculated to be 509.6, 437.6, 126.6, and 268.7 Tg $season^{-1}$ for December–January–February (DJF), March–April–May (MAM), June–July–August (JJA), and September–October–November (SON), respectively. Although the seasonal flux was highest in DJF, the monthly $O_3$ flux through the panel peaked in March and reached the nadir in July (not shown in Fig. 6). Such monthly variations of the Asian $O_3$ outflow flux agreed with those in Liu et al. (2002). The maximum $O_3$ outflow in March was caused by the combined effects of meteorological conditions, biomass burning emissions, and stratospheric $O_3$ intrusion. The "warm conveyor belt" (WCB) airstreams that export pollution from the Asian boundary layer to the free troposphere, and the mid-latitude prevailing westerly winds in the free troposphere that transport pollution from Asia to the Northwest Pacific, were strongest during the early spring period (Eckhardt et al., 2004; Pochanart et al., 2004). The contribution from Asian biomass burning emissions on $O_3$ outflow was maximum in the spring and insignificant during other seasons (Liu et al., 2002). The stratospheric $O_3$ intrusion was also found to be most effective in late winter and early spring (Danielsen and Mohnen, 1977; Mahlman and

Moxim, 1978).

## 4.2 IAVs and decadal trends of Asian $O_3$ outflow

Figure 7a shows the simulated annual $O_3$ outflow fluxes through the meridional plane along $135\,°E$ from $20\,°N$ to $55\,°N$, from the surface to 100 hPa, during 1986–2006 in the Met, Emis, and MetEmis simulations, and Fig. 7b shows the associated deviations from the mean (DEV). The simulations of the $O_3$ outflow in Met, Emis, and MetEmis examined, respectively, the effects of variations in meteorological parameters alone, anthropogenic emissions alone, and both meteorological parameters and anthropogenic emissions. The outflow fluxes of $O_3$ with changes in anthropogenic emissions alone (the Emis simulation) exhibited a statistically significant ($P < 0.001$) increasing trend. However, the magnitude of the increasing trend was very small; the decadal trend of the Asian $O_3$ outflow flux in the Emis simulation was calculated to be $+16.7$ Tg decade$^{-1}$ (i.e., $+1.2$ % decade$^{-1}$) using the linear fit with least-squares method. The DEV, defined as

$$\text{DEV} = 100\% \times \left( C_i - \frac{1}{n}\sum_{i=1}^{n} C_i \right) \Big/ \left( \frac{1}{n}\sum_{i=1}^{n} C_i \right),$$

where $n$ is the number of years examined ($n = 21$ for 1986–2006) and $C_i$ is the simulated $O_3$ outflow flux in year $i$, changed from $-1.3$ % (in 1986) to $+1.4$ % (in 2006), also indicating a small increasing trend in the $O_3$ outflow flux. With variations in meteorological parameters alone (the Met simulation), simulated $O_3$ outflow fluxes exhibited large IAVs, but a statistically insignificant ($P > 0.05$) decadal trend of $-3.4$ % decade$^{-1}$. The DEV values in the Met simulation ranged from $-8$ % to $+16.5$ %. With variations in both anthropogenic emissions and meteorological parameters (the MetEmis simulation), the simulated $O_3$ outflow showed large IAVs, but a statistically insignificant ($P > 0.05$) decadal trend of $-2.2$ % decade$^{-1}$.

To analyze the IAVs of $O_3$ outflow fluxes, the decadal trend obtained from the linear fit was removed from the time series of simulated $O_3$ outflow fluxes, following the approach used in previous studies that examined IAVs of aerosol outflow fluxes (Yang et al., 2015) and $O_3$ concentrations (Camp et al., 2003). The deviations from the mean of the detrended $O_3$ outflow fluxes from the Met, Emis, and MetEmis simulations over 1986–2006 are shown in Fig. 7c. While the detrended outflow fluxes of $O_3$ in Met and MetEmis simulations showed large IAVs with DEV values in the range of $-7.5$ % to $+13.5$ %, the DEV values in the Emis simulation were very small (in the range of $\pm 0.3$ %). The two deviation curves from the Met and MetEmis simulations almost coincided with each other, indicating the dominant role of variations in meteorological parameters in the IAVs of the Asian $O_3$ outflow flux.

The IAVs in the $O_3$ outflow fluxes were further quantified with statistical variables of mean absolute deviation (MAD) and absolute percent departure from the mean (APDM), which have been used in previous IAV studies, such

as Mu and Liao (2014), Lou et al. (2015), and Yang et al. (2015). The absolute IAVs of the $O_3$ outflow fluxes can be quantified by MAD, defined as

$$\mathrm{MAD} = \frac{1}{n}\sum_{i=1}^{n}|C_i - \frac{1}{n}\sum_{i=1}^{n}C_i|,$$

while the IAVs relative to the multi-year average outflow flux can be quantified by APDM, defined as

$$\mathrm{APDM} = 100\% \times \mathrm{MAD}/\left(\frac{1}{n}\sum_{i=1}^{n}C_i\right),$$

where $n$ is the number of years examined ($n = 21$ for years 1986–2006) and $C_i$ is the detrended $O_3$ outflow flux in year $i$. The MAD and APDM values of the detrended seasonal and annual $O_3$ outflow fluxes across the meridional plane along 135 °E from 20 °N to 55 °N, from the surface to 100 hPa, are shown in Fig. 8. The seasonal MAD and APDM values in the Emis simulation were close to zero, while those in the Met and MetEmis simulations were relatively large. The APDM values in the Met and MetEmis simulations were maximum in JJA and minimum in SON. The MAD and APDM values in the Met simulation were almost identical to those in the MetEmis simulation, which indicated again that the IAVs of the $O_3$ outflow fluxes were mainly dependent on the variations in meteorological conditions, rather than the variations in anthropogenic emissions. With variations in both meteorological parameters and anthropogenic emissions, the seasonal APDM values were in the range of 4–9 % and the annual APDM value was 3.3 %.

Figure 9 shows the pressure–latitude cross-sections of MAD values along 135 °E for detrended annual $O_3$ outflow fluxes from the Met, Emis, and MetEmis simulations. The $O_3$ outflow in the Met simulation exhibited large IAVs throughout the whole troposphere, with MAD values greater than 0.2 kg yr$^{-1}$ m$^{-2}$. The MAD values increased with altitude, which could be attributed to the vertical distributions of the IAVs in westerly winds (see MAD values of winds in Fig. 9a). The variations in anthropogenic emissions led to very small IAVs, with MAD values less than 0.2 kg yr$^{-1}$ m$^{-2}$ (Fig. 9b) throughout the troposphere. With both variations in meteorological parameters and anthropogenic emissions, the MAD values (Fig. 9c) showed almost identical magnitudes and spatial distributions to those in the Met simulation (Fig. 9a), indicating the dominant role of variations in meteorological conditions in the IAVs of the $O_3$ outflow.

Variations in meteorological conditions can influence the IAVs of the $O_3$ outflow fluxes by changing $O_3$ concentrations over East Asia (Yang et al., 2014; Lou et al., 2015), and by altering zonal winds (Kurokawa et al., 2009). The $O_3$ outflow flux is simulated to correlate positively with zonal wind averaged over 20 °–55 °N along 135 °E, with a high correlation coefficient of +0.71 for annual fluxes and zonal winds. The correlation coefficient between $O_3$ fluxes and zonal winds is calculated to be +0.96 during summer when the APDM values of $O_3$ outflow fluxes are maximum. The high correlation coefficients indicate that the variation in zonal winds is the key factor that leads to the large IAVs of $O_3$ outflow fluxes.

## 4.3 Effect of variations in biomass burning emissions

The biomass burning emissions of the $O_3$ precursors exhibited large IAVs during 1997–2006 (Fig. 1). To analyze the impacts of biomass burning emissions on IAVs of $O_3$ outflow fluxes, we compare the MAD and APDM values of detrended $O_3$ outflow fluxes during 1997–2006 in the MetEmis and MetEmisB simulations. The MAD (APDM) was calculated to be 31.17 Tg $yr^{-1}$ (2.35 %) in the MetEmis simulation and 31.82 Tg $yr^{-1}$ (2.36 %) in the MetEmisB simulation. The minor influence of biomass burning emissions on the IAVs of the $O_3$ outflow fluxes from East Asia was also supported by Voulgarakis et al. (2015). Furthermore, Lin et al. (2014) reported that meteorological variability, compared with the variability in biomass burning, was much more important for driving the IAVs in springtime $O_3$ at the Mauna Loa Observatory, a remote North Pacific site sensitive to Asian pollution outflow.

## 5 Future changes in Asian $O_3$ outflow for 2000–2050

In this part of the study, we quantify future decadal changes in Asian $O_3$ outflow during 2000–2050 under the SRES A1B scenario, and examine the relative impacts of variations in climate and anthropogenic emissions on the changes. We conduct each simulation for 10 years, driven by 1996–2005 meteorology to represent the present-day (year 2000) climate, and by 2046–2055 meteorological fields to represent the future (year 2050) climate, following 1 year of model spin-up. All the results presented below are 10-year averages. Simulated present-day and future changes in seasonal and annual fluxes of $O_3$ across the vertical plane along 135 °E from 20 °N to 55 °N are summarized in Table 5.

## 5.1 Present-day $O_3$ outflow

The pressure–latitude cross-sections along 135 °E of the simulated present-day (Met2000Emis2000) seasonal $O_3$ outflow fluxes, driven by the meteorological inputs provided by GISS GCM 3, are shown in Fig. 10a. The magnitudes, spatial distributions, and seasonal variations agree closely with those driven by the assimilated GEOS-4 meteorological fields (Fig. 6). The $O_3$ outflow flux through the vertical plane is simulated to be 1877.1 Tg $yr^{-1}$ with GISS GCM 3 meteorology, and 1342.5 Tg $yr^{-1}$ with the GEOS-4 assimilated meteorological fields, which indicates the reliability of the simulated present-day $O_3$ outflow fluxes.

## 5.2 Effect of future changes in climate alone

Relative to the present-day value, year 2050 annual outflow of $O_3$ is estimated to increase by 2.0 % (Table 5) as a

result of climate change alone (Met2050Emis2000 minus Met2000Emis2000). The outflow of $O_3$ shows a slight decrease of 1.8 % in DJF and of 3.8 % in SON, but a large increase of 14.5 % in JJA and of 7.3 % in MAM. The spatial distributions of projected changes in $O_3$ fluxes are well consistent with those of changes in zonal winds (Fig. 10b). The wind speed of the westerlies in DJF and SON decreases across the troposphere over 30–45 °N, leading to the reductions in the $O_3$ outflow fluxes. In contrast, the increases in zonal winds in JJA and MAM lead to the increases of $O_3$ outflow fluxes throughout the troposphere over 30–45 ° N. Our projected future changes in zonal winds are consistent with previous studies. By analyzing 18 CMIP5 models, Huang and Wang (2016) assessed the future changes in atmospheric circulation during spring over East Asia. They found that, although different models projected different changes (even in sign) in zonal winds, the ensemble mean of five better-skill models among the 18 CMIP5 models exhibited overall increases in zonal winds throughout the whole troposphere during spring, which agrees with our simulation. Based on 31 (29)-model ensemble mean results, Jiang and Tian (2013) showed that the westerlies along 135 °E during winter (summer) were projected to weaken (strengthen). Such projected patterns of future changes in westerlies during winter and summer are also captured by our model. Changes in $O_3$ concentrations also contribute to the changes in $O_3$ outflow; although the zonal winds are projected to increase north of 40 °N in the upper troposphere during SON, the $O_3$ outflow fluxes are simulated to decrease because of the significant decreases of $O_3$ levels north of 40 °N in the upper troposphere (Fig. S1).

### 5.3 Effect of future changes in anthropogenic emissions alone

The annual outflow of $O_3$ through the vertical plane is simulated to increase by 7.9 % relative to the present-day value (Table 5) as a result of the changes in anthropogenic emissions alone (Met2000Emis2050 minus Met2000Emis2000). Considering that the $O_3$ outflow with changes in anthropogenic emissions alone exhibits an increasing trend of 1.2 % decade$^{-1}$ over 1986–2006 (Sect. 4.2), the increasing trend of 1.2 % decade$^{-1}$ (i.e., 6.0 % half-century$^{-1}$) is close to the value of 7.9 % over the future half-century.

The projected future $O_3$ fluxes show increases during all seasons, which can be attributed to the increases in $O_3$ concentrations at all altitudes over Asia and upwind regions (i.e., Europe and Central Asia; Fig. S1) as a result of the increases in anthropogenic emissions of the $O_3$ precursors ($NO_x$ and NMVOCs) and $CH_4$ concentrations. $NO_x$ emissions in 2050 are projected to increase by 159.6% over Asia and by 78.4% globally, while NMVOCs emissions are projected to increase by 118.6% over Asia and by 89.4% globally under the SRES A1B scenario (Table 1). The $CH_4$ mixing ratios are projected to increase by 37.1% relative to the present-day value. The largest increases of $O_3$ outflow fluxes are located in the middle-upper troposphere (Fig. 10c) owing to the strong westerlies located here. It is

noted that, in spite of the significant increases of emissions, the simulated surface-layer $O_3$ concentrations show slight decreases over the North China Plain in DJF, which subsequently leads to the small decreases of $O_3$ outflow fluxes at the surface layer over 30–40 °N. In DJF, biogenic VOC emissions are especially low over the North China Plain, whereas anthropogenic $NO_x$ emissions are fairly high due to the residential heating, leading to a low VOCs/$NO_x$ ratio in this region (Lou et al., 2010; Fu et al., 2012). Therefore, increases in $NO_x$ emissions lead to decreases in surface-layer $O_3$ concentrations over the North China Plain.

### 5.4 Effect of future changes in both climate and anthropogenic emissions

The annual outflow of $O_3$ through the vertical plane is projected to increase by 12.2 % (Table 5) during 2000–2050 as a consequence of future changes in both climate and anthropogenic emissions (Met2050Emis2050 minus Met2000Emis2000). Climate change in DJF and SON slightly offsets the effects of changes in anthropogenic emissions, while climate change in MAM and JJA enhances the effects of variations in anthropogenic emissions. When considering future changes in both emissions and climate, the projected $O_3$ outflow fluxes show increases throughout almost the entire troposphere along 135 °E during all seasons (Fig. 10d).

### 6 Uncertainty discussion

There are some uncertainties in our simulations. First, the influence of interannual variation in stratosphere-troposphere exchange on tropospheric $O_3$ is not considered in this study. Terao et al. (2008) reported that the stratosphere-troposphere exchange had large impacts on the interannual variability of tropospheric $O_3$ over Canada and Europe but the impact was much smaller over East Asia. The second is the uncertainty associated with the selection of longitudinal transect. We calculate $O_3$ flux through the vertical plane along 135 °E, because 135 °E is the easternmost boundary of China (i.e., Wusuli River in Northeastern China). We also calculate the $O_3$ outflow flux along 120 °E, more close to ozone production region in central-eastern China, and find that the variations in $O_3$ fluxes calculated at 120 °E are similar to those calculated at 135 °E. With variations in both anthropogenic emissions and meteorological parameters (the MetEmis simulation), the simulated $O_3$ outflow shows large IAVs but a statistically insignificant ($P > 0.05$) trend. The conclusion is consistent with that drawn from the variations in $O_3$ outflow calculated at 135 °E. Finally, projecting future atmospheric circulation on regional scales has large uncertainty, which is undergoing continuing improvement.

### 7 Conclusions

We quantify the past and future changes in the $O_3$ outflow from East Asia using the global 3-D chemical transport model GEOS-Chem. The historical (1986–2006) simulations are driven by the assimilated GEOS-4 meteorological fields, and the future (2000–2050) simulations under the IPCC SRES A1B scenario are driven by the meteorological fields archived from GISS GCM 3. Sensitivity studies are conducted to examine the respective impacts of meteorological parameters and emissions on the variations in the outflow flux.

The measurements from WDCGG and EANET are used to evaluate the simulated surface-layer $O_3$ concentrations; the ozonesonde data from WOUDC are used to evaluate the simulated $O_3$ concentrations for the boundary layer, middle and upper troposphere. Generally, the seasonal cycles and interannual variations of tropospheric $O_3$ concentrations are captured fairly well by the GEOS-Chem model, with high correlation coefficients of 0.82–0.93 at four ground-based sites and 0.55–0.88 at two ozonesonde sites. The increasing trends in surface-layer $O_3$ concentrations in East Asia over the past two decades can also be captured by the GEOS-Chem model, although the modeled $O_3$ trends have low biases. Simulated Asian $O_3$ outflow flux peaks in early spring, and reaches the nadir in summer. The maximum $O_3$ fluxes are located in the middle-upper troposphere.

The IAVs and decadal trends of Asian $O_3$ outflow are examined over 1986–2006. Simulated $O_3$ outflow fluxes showed large IAVs, but an insignificant decadal trend; with variations in both meteorological parameters and anthropogenic emissions, the seasonal APDM values were in the range of 4–9 %. Sensitivity simulations showed that the large IAVs of the $O_3$ outflow fluxes were mainly caused by the variations in meteorological conditions, rather than the variations in anthropogenic and biomass burning emissions. Although variations in meteorological parameters could influence the IAVs of the $O_3$ outflow fluxes by changing $O_3$ concentrations over East Asia and by altering zonal winds, the latter was identified to be the key factor because of the high correlation coefficient of +0.71 between the annual fluxes and zonal winds.

The decadal changes in Asian $O_3$ outflow are also examined during 2000–2050. The present-day annual $O_3$ flux through the vertical plane is calculated as 1877.1 Tg, which is projected to change over 2000–2050 by +2.0 %, +7.9 %, and +12.2 %, respectively, due to climate change alone, emissions change alone, and changes in both climate and emissions. During MAM and JJA, climate change plays a larger role in the future changes in $O_3$ outflow compared with emissions changes, owing to the significant increases in zonal winds during these two seasons. It is noted that climate change will aggravate the impacts of increases in anthropogenic emissions on the $O_3$ outflow from East Asia over 2000–2050 under the SRES A1B scenario.

These findings are helpful for understanding the temporal evolutions of tropospheric $O_3$ on different timescales in the downwind regions of East Asia. Observed IAVs of tropospheric $O_3$ on a relatively short timescale can be attributed to variations in meteorological parameters. Furthermore, conclusions from this study will have important implications

for long-term air quality planning for the downwind regions of China, such as Japan and US. Since future climate change will increase $O_3$ outflow from East Asia, extra efforts are needed to reduce anthropogenic emissions of $O_3$ precursors to offset the adverse effects caused by climate change.

**8 Data availability**

GEOS-Chem is an open-access model developed collaboratively at Harvard University and other institutes in North America, Europe, and Asia. The source codes can be downloaded from http://acmg.seas.harvard.edu/geos/. The tropospheric $NO_2$ vertical column density (VCD) data are retrieved from GOME (1996–2002) and SCIAMACHY (2003–2006), which are available from www.temis.nl. The $O_3$ measurements at Minamitorishima and Yonagunijima are available from the World Data Centre for Greenhouse Gases (WDCGG, www.ds.data.jma.go.jp/gmd/wdcgg/), and those at Rishiri and Ogasawara are available from the Acid Deposition Monitoring Network in East Asia (EANET, www.eanet.asia/product/index.html). The ozonesonde data at Naha and Tsukuba are available from the World Ozone and Ultraviolet Radiation Data Centre (WOUDC, www.woudc.org). All data presented in this study are available upon request from the corresponding author.

*Author contributions.* H. Liao and J. Zhu conceived the study and designed the experiments. J. Zhu performed the simulations, carried out the data analysis, and prepared the manuscript. Y. Mao provided useful comments on the paper. Y. Yang and H. Jiang helped with performing the experiments.

*Competing interests.* The authors declare that they have no conflicts of interest.

*Acknowledgements.* This work was supported by the National Basic Research Program of China (973 program, Grant No. 2014CB441202) and the National Natural Science Foundation of China under grants 91544219 and 41475137. We acknowledge the free use of GOME and SCIAMACHY tropospheric $NO_2$ vertical column density (VCD) data available from www.temis.nl. The following data centers are also acknowledged: the World Data Centre for Greenhouse Gases (WDCGG, www.ds.data.jma.go.jp/gmd/wdcgg/) operated by Japan Meteorological Agency (JMA) in cooperation with World Meteorological Organization (WMO); the World Ozone and Ultraviolet Radiation Data Centre (WOUDC, www.woudc.org) operated by Environment Canada for the Global Atmosphere Watch (GAW) program of WMO. The Rishiri and Ogasawara sites are operated by Ministry of the Environment of Japan as part of the Acid Deposition Monitoring Network in East Asia (EANET, www.eanet.asia/product/index.html) program. We are also very grateful to the reviewers for their helpful comments and thoughtful suggestions.

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

**Table 1.** Anthropogenic emissions[a] of $O_3$ precursors for the present day (year 2000) and future (year 2050, under the SRES A1B scenario).

| Species | Global | | | Asian[b] | | |
|---|---|---|---|---|---|---|
| | 2000 | 2050 | Change (%) | 2000 | 2050 | Change (%) |
| $NO_x$ (Tg N yr$^{-1}$) | 33.4 | 59.6 | +78.4 | 10.9 | 28.3 | +159.6 |
| CO (Tg CO yr$^{-1}$) | 1054.2 | 1332.0 | +26.4 | 393.7 | 487.2 | +23.7 |
| NMVOCs (Tg C yr$^{-1}$) | 70.8 | 134.1 | +89.4 | 28.5 | 62.3 | +118.6 |
| $CH_4$ (ppbv) | 1750 | 2400 | +37.1 | 1750 | 2400 | +37.1 |

[a] Biomass burning emissions are included.

[b] Asia covers the domain of 60–150 °E, 10 °S–55 °N.

**Table 2.** Natural emissions of $O_3$ precursors for the present day (year 2000) and future (year 2050, under the SRES A1B scenario).

| Species | Global | | | Asian[a] | | |
|---|---|---|---|---|---|---|
| | 2000 | 2050 | Change (%) | 2000 | 2050 | Change (%) |
| Lightning $NO_x$ (Tg N yr$^{-1}$) | 4.8 | 5.7 | +18.8 | 1.2 | 1.4 | +16.7 |
| Soil $NO_x$ (Tg N yr$^{-1}$) | 6.7 | 7.7 | +14.9 | 1.4 | 1.7 | +21.4 |
| Biogenic VOCs (Tg C yr$^{-1}$) | 614.5 | 750.2 | +22.1 | 106.1 | 126.2 | +18.9 |

[a] Asia covers the domain of 60–150 °E, 10 °S–55 °N.

**Table 3.** Experimental design of the simulations for 1986–2006.

| Simulation | Simulated years | Meteorological parameters | Anthropogenic emissions | $CH_4$ abundance | Biomass burning emissions |
|---|---|---|---|---|---|
| Met | 1986–2006 | Vary over 1986–2006 | Fixed at 2006 | Fixed at 2006 | Turn off |
| Emis | 1986–2006 | Fixed at 2006 | Vary over 1986–2006 | Vary over 1986–2006 | Turn off |
| MetEmis | 1986–2006 | Vary over 1986–2006 | Vary over 1986–2006 | Vary over 1986–2006 | Turn off |
| MetEmisB[a] | 1997–2006 | Vary over 1997–2006 | Vary over 1997–2006 | Vary over 1997–2006 | Vary over 1997–2006 |

[a] The MetEmisB simulation is conducted for 1997–2006 owing to the unavailability of biomass burning emissions before 1997.

**Table 4.** Information for the sites with $O_3$ measurements used in model evaluation.

| Site | Location | Database | Height | R[a] | NMB[b] (%) |
|---|---|---|---|---|---|
| Minamitorishima | 24.3 °N, 154.0 °E | WDCGG | surface | 0.92 | +12.7 |
| Yonagunijima | 24.5 °N, 123.0 °E | WDCGG | surface | 0.93 | +12.6 |
| Rishiri | 45.1 °N, 141.2 °E | EANET | surface | 0.82 | +2.4 |
| Ogasawara | 27.1 °N, 142.2 °E | EANET | surface | 0.90 | +29.6 |
| Naha | 26.2 °N, 127.7 °E | WOUDC | 500–300 hPa | 0.68 | −2.61 |
| | | | 700–500 hPa | 0.77 | +16.4 |
| | | | 850–700 hPa | 0.85 | +24.3 |
| | | | 1000–850 hPa | 0.88 | +39.5 |
| Tsukuba | 36.1 °N, 140.1 °E | WOUDC | 500–300 hPa | 0.55 | +15.8 |
| | | | 700–500 hPa | 0.76 | +12.3 |
| | | | 850–700 hPa | 0.76 | +8.61 |
| | | | 1000–850 hPa | 0.60 | +8.5 |

[a] Correlation coefficient (R) between the observed and simulated monthly $O_3$ mixing ratios.

[b] Normalized mean bias (NMB, %) between the observed and simulated monthly $O_3$ mixing ratios.

**Table 5.** Simulated present-day O$_3$ fluxes and projected changes from the present day (1996–2005) to the future (2046–2055) through the vertical plane along 135 °E from 20 °N to 55 °N, and from the surface to 100 hPa, due to future climate change alone, change in anthropogenic emissions alone, and changes in both climate and emissions.

| | O$_3$ Fluxes[a] | | | |
|---|---|---|---|---|
| | Met2000Emis2000 (present day) | Met2050Emis2000 (climate) | Met2000Emis2050 (emissions) | Met2050Emis2050 (climate + emissions) |
| DJF | 792.9 | 778.3 (−1.8%) | 850.5 (+7.3%) | 853.7 (+7.7%) |
| MAM | 597.0 | 640.4 (+7.3%) | 639.0 (+7.0%) | 698.0 (+16.9%) |
| JJA | 146.5 | 167.7 (+14.5%) | 161.3 (+10.1%) | 187.2 (+27.8%) |
| SON | 340.7 | 327.8 (−3.8%) | 374.1 (+9.8%) | 368.1 (+8.0%) |
| Annual | 1877.1 | 1914.1 (+2.0%) | 2024.9 (+7.9%) | 2106.9 (+12.2%) |

[a] The units are Tg season$^{-1}$ for seasonal fluxes and Tg yr$^{-1}$ for annual fluxes. Values in parentheses are percentage changes relative to the present-day fluxes.

**Figure captions**

**Figure 1.** Evolution of annual anthropogenic and biomass burning emissions summed over the globe and Asia (60–150 ° E, 10 ° S–55 ° N) for $NO_x$ (Tg N $yr^{-1}$), CO (Tg CO $yr^{-1}$), and NMVOCs (Tg C $yr^{-1}$) over 1986–2006. Blue squares represent anthropogenic emissions, and red circles represent the sum of anthropogenic emissions and biomass burning emissions. The last panel shows the evolution of global $CH_4$ abundance (ppbv) during 1986–2006.

**Figure 2.** Evolution of annual natural emissions summed over the globe and Asia (60–150 ° E, 10 ° S–55 ° N) for lightning $NO_x$ (Tg N $yr^{-1}$), soil $NO_x$ (Tg N $yr^{-1}$), and biogenic VOCs (Tg C $yr^{-1}$) over 1986–2006.

**Figure 3.** Time series of monthly surface-layer $O_3$ mixing ratios measured by WDCGG and EANET (blue line), and simulated by MetEmisB (red line). (a) Minamitorishima and (b) Yonagunijima are WDCGG sites, and (c) Rishiri and (d) Ogasawara sites are EANET sites.

**Figure 4.** Time series of monthly $O_3$ mixing ratios measured by ozonesonde (blue line), and simulated by MetEmisB (red line). (a) Naha and (b) Tsukuba are ozonesonde sites from WOUDC. Comparisons are shown for four altitude levels in the troposphere.

**Figure 5**. Comparison of simulated trends in seasonal or annual mean surface-layer $O_3$ concentrations from the MetEmis experiment with observations for Hongkong (location: 22.2 °N, 114.3 °E; years: 1994–2007; reference: Wang et al., 2009b), Waliguan (36.3 °N, 100.9 °E; 1994–2013; Xu et al. 2016), Beijing (40.0 °N, 116.5 °E; 2001–2006; Tang et al., 2009), and Taiwan (23.5 °N, 121.0 °E; 1994–2007; Lin et al., 2010). The simulated trend at Waliguan site for winter is statistically insignificant. The trends in seasonal-mean $O_3$ concentrations at Taiwan station are unavailable.

**Figure 6.** The pressure–latitude cross-sections along 135 ° E of the simulated seasonal $O_3$ outflow fluxes and zonal winds during four seasons averaged over 1997–2006 in the MetEmisB simulation. The $O_3$ mass fluxes are shown by shades (units: kg $season^{-1}$ $m^{-2}$), and winds are represented by contours (units: m $s^{-1}$). Positive fluxes represent eastward fluxes, and negative values represent westward fluxes.

**Figure 7.** Evolution of (a) annual $O_3$ outflow fluxes (Tg $yr^{-1}$) across the meridional plane along 135 °E from 20 °N to 55 ° N, and from the surface to 100 hPa, over 1986–2006 in the Met, Emis, and MetEmis simulations; (b) the associated deviations from the mean (%); and (c) deviations from the mean (%) of the detrended $O_3$ outflow fluxes. The deviation from the mean (DEV) is defined in Sect. 4.2.

**Figure 8.** The MAD and APDM values of the detrended seasonal and annual $O_3$ outflow fluxes across the meridional plane along 135 °E from 20 °N to 55 °N, and from the surface to 100 hPa, over 1986–2006 in Met, Emis, and MetEmis simulations. Both the MAD and APDM are defined in Sect. 4.2. The units of MAD are Tg $season^{-1}$ for seasonal fluxes and Tg $yr^{-1}$ for annual fluxes.

**Figure 9.** The pressure–latitude cross-sections along 135 ° E of MAD values for detrended annual $O_3$ outflow fluxes and zonal winds over 1986–2006 in the Met, Emis, and MetEmis simulations. The MAD values for $O_3$ mass fluxes are shown by shades (units: kg $yr^{-1}$ $m^{-2}$), and the MAD values for winds are represented by contours (units: m $s^{-1}$).

**Figure 10.** (a) The pressure–latitude cross-sections along 135 ° E of simulated present-day $O_3$ mass fluxes and zonal winds (Met2000Emis2000). Projected changes in $O_3$ mass fluxes and zonal winds from the present day (1996–2005) to the future (2046–2055) caused by (b) climate change alone (Met2050Emis2000 minus Met2000Emis2000); (c) changes in anthropogenic emissions alone (Met2000Emis2050 minus Met2000Emis2000); and (d) changes in both

climate and anthropogenic emissions (Met2050Emis2050 minus Met2000Emis2000). The $O_3$ mass fluxes are shown by shades (units: kg $season^{-1}$ $m^{-2}$), and winds are represented by contours (units: m $s^{-1}$). The dotted areas are statistically significant at the 95 % level, as determined by a two-sample Student's *t*-test.

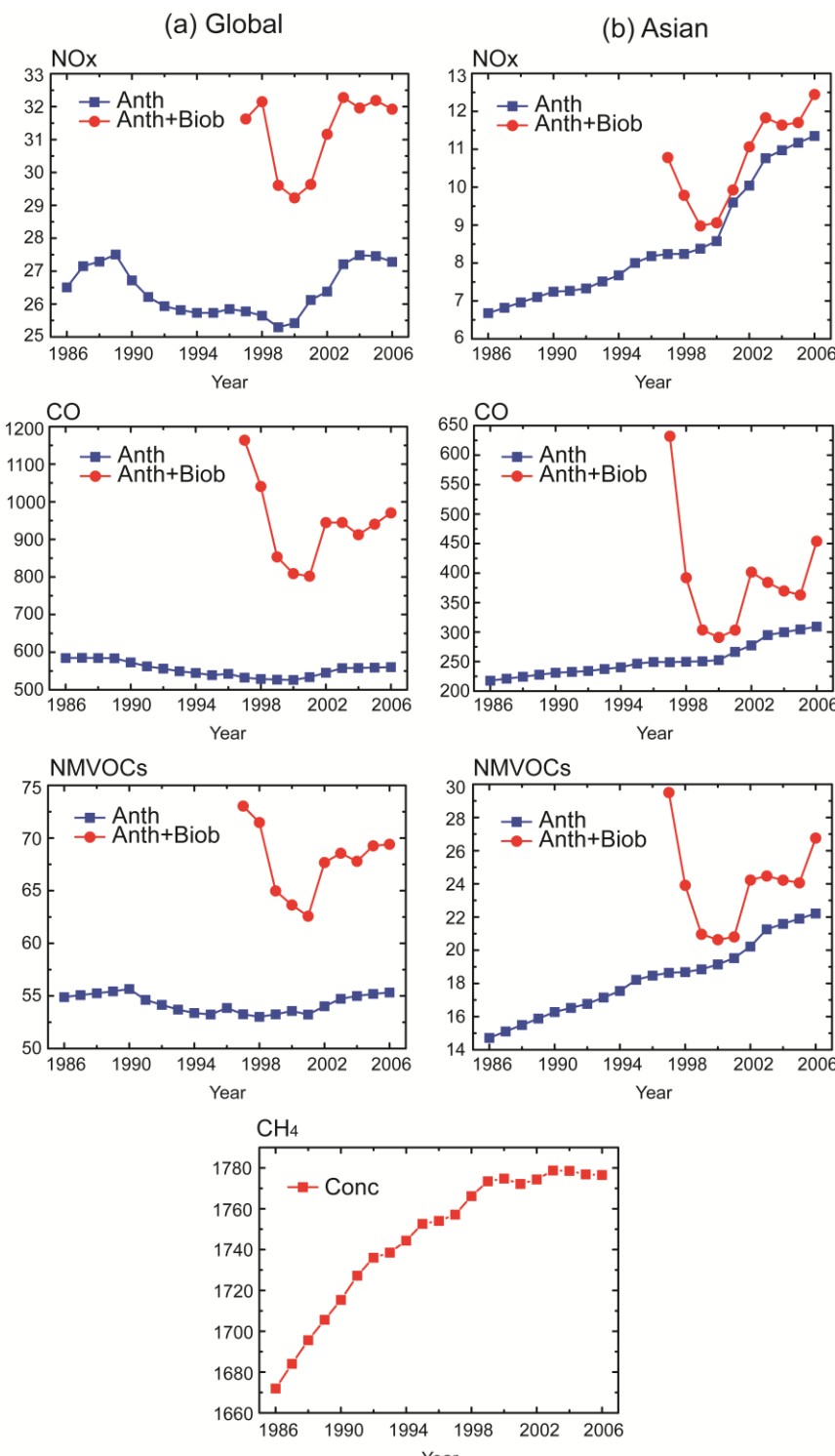

**Figure 1.** Evolution of annual anthropogenic and biomass burning emissions summed over the globe and Asia (60–150 °E, 10 °S–55 °N) for $NO_x$ (Tg N $yr^{-1}$), CO (Tg CO $yr^{-1}$), and NMVOCs (Tg C $yr^{-1}$) over 1986–2006. Blue squares represent anthropogenic emissions, and red circles represent the sum of anthropogenic emissions and biomass burning emissions. The last panel shows the evolution of global $CH_4$ abundance (ppbv) during 1986–2006.

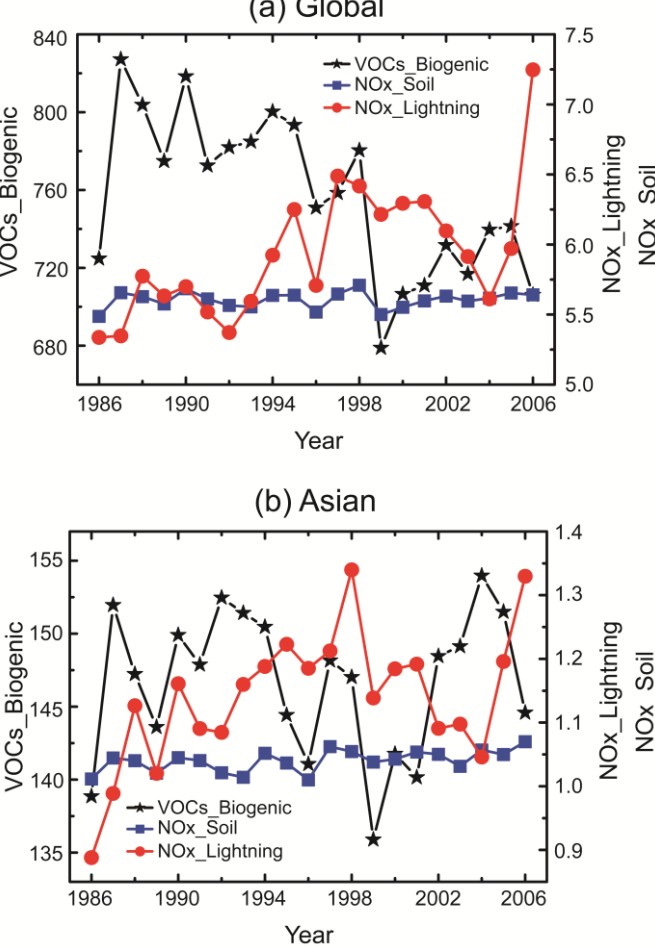

**Figure 2.** Evolution of annual natural emissions summed over the globe and Asia (60–150 ° E, 10 ° S–55 ° N) for lightning NO$_x$ (Tg N yr$^{-1}$), soil NO$_x$ (Tg N yr$^{-1}$), and biogenic VOCs (Tg C yr$^{-1}$) over 1986–2006.

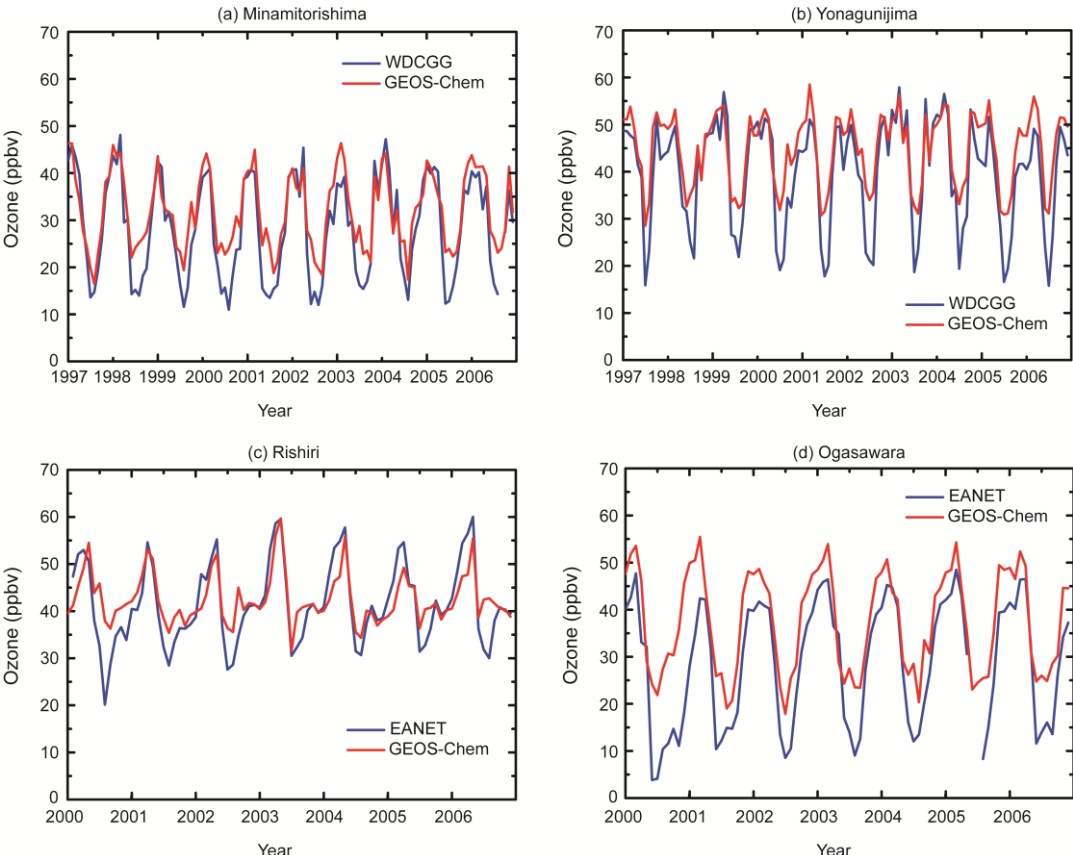

**Figure 3.** Time series of monthly surface-layer O$_3$ mixing ratios measured by WDCGG and EANET (blue line), and simulated by MetEmisB (red line). (a) Minamitorishima and (b) Yonagunijima are WDCGG sites, and (c) Rishiri and (d) Ogasawara sites are EANET sites.

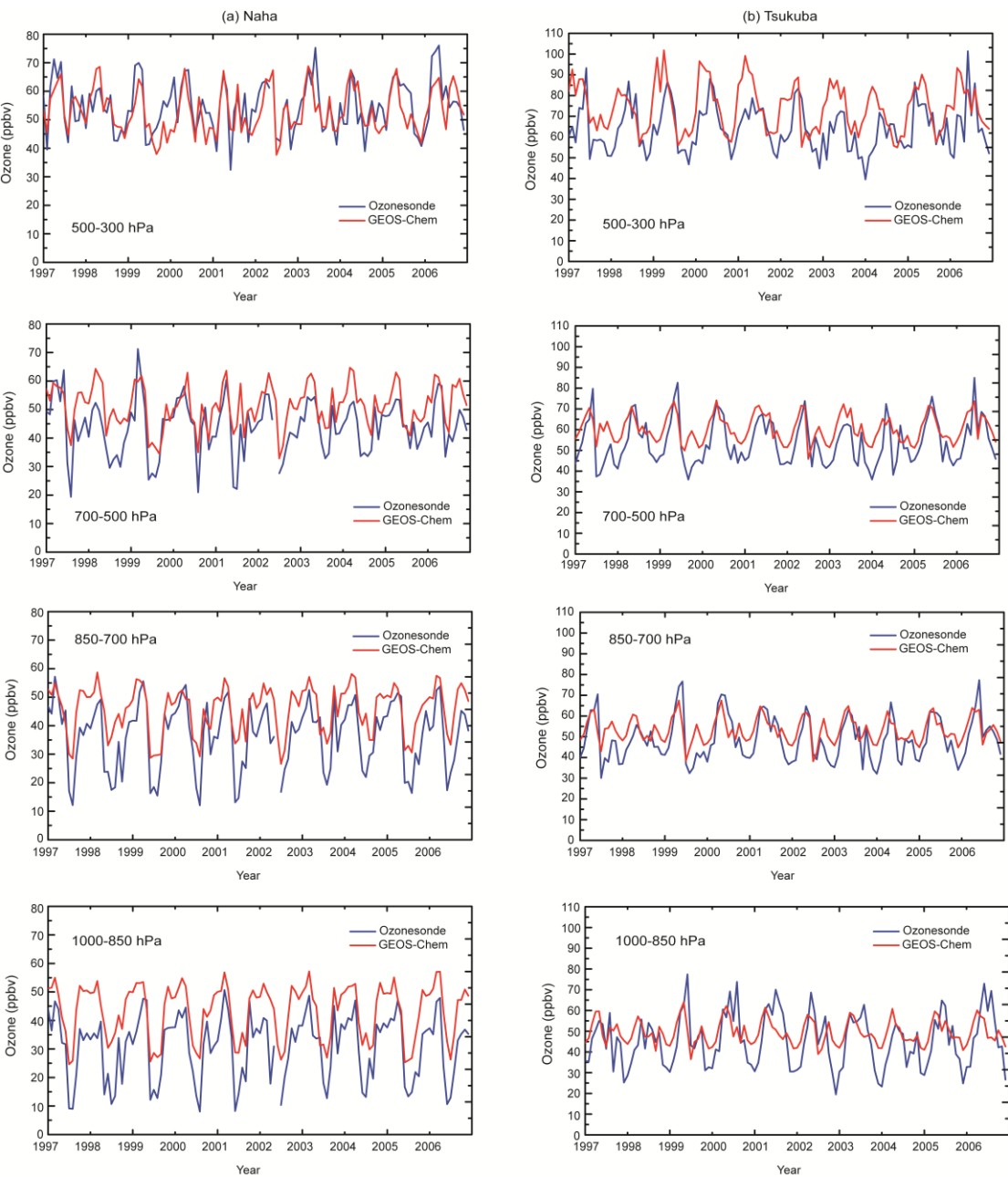

**Figure 4.** Time series of monthly O$_3$ mixing ratios measured by ozonesonde (blue line), and simulated by MetEmisB (red line). (a) Naha and (b) Tsukuba are ozonesonde sites from WOUDC. Comparisons are shown for four altitude levels in the troposphere.

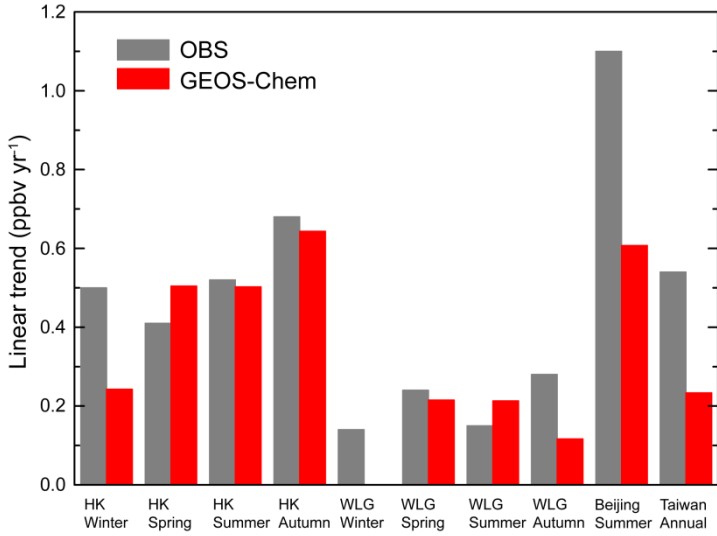

**Figure 5**. Comparison of simulated trends in seasonal or annual mean surface-layer $O_3$ concentrations from the MetEmis experiment with observations for Hongkong (location: 22.2 ° N, 114.3 ° E; years: 1994–2007; reference: Wang et al., 2009b), Waliguan (36.3 °N, 100.9 °E; 1994–2013; Xu et al. 2016), Beijing (40.0 °N, 116.5 °E; 2001–2006; Tang et al., 2009), and Taiwan (23.5 °N, 121.0 °E; 1994–2007; Lin et al., 2010). The simulated trend at Waliguan site for winter is statistically insignificant. The trends in seasonal-mean $O_3$ concentrations at Taiwan station are unavailable.

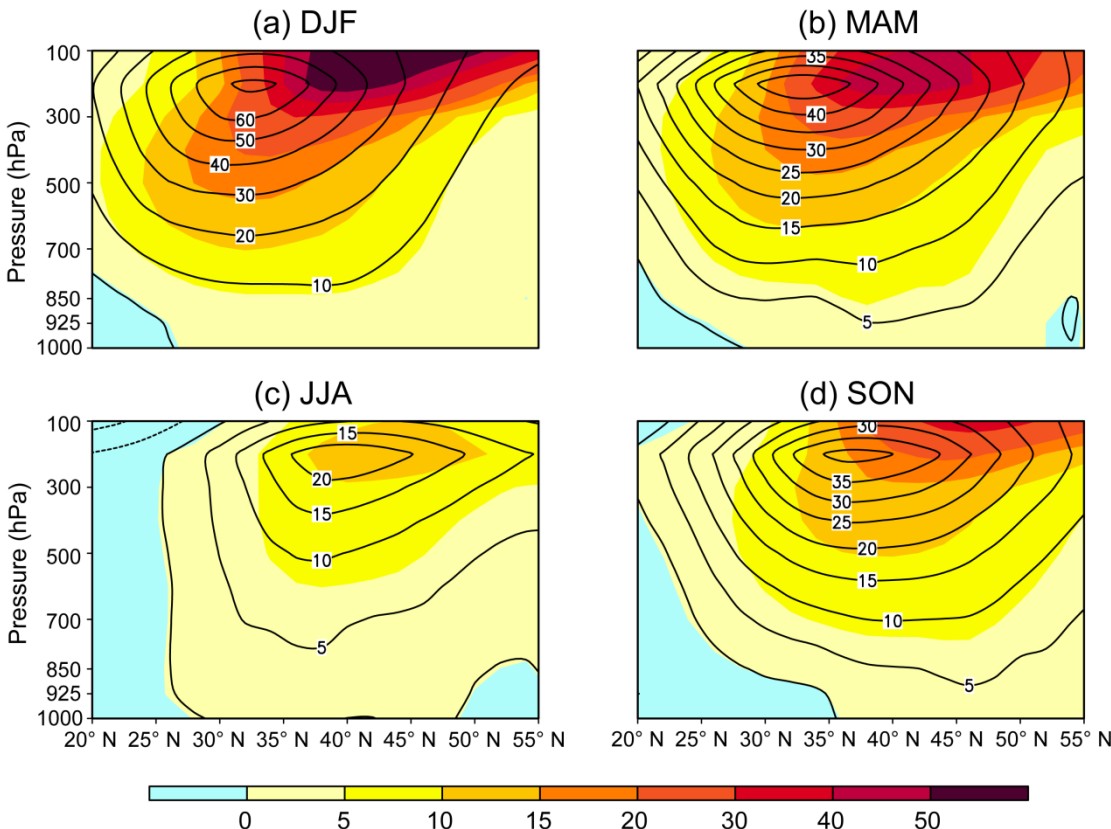

**Figure 6.** The pressure–latitude cross-sections along $135\,^{\circ}\,E$ of the simulated seasonal $O_3$ outflow fluxes and zonal winds during four seasons averaged over 1997–2006 in the MetEmisB simulation. The $O_3$ mass fluxes are shown by shades (units: kg season$^{-1}$ m$^{-2}$), and winds are represented by contours (units: m s$^{-1}$). Positive fluxes represent eastward fluxes, and negative values represent westward fluxes.

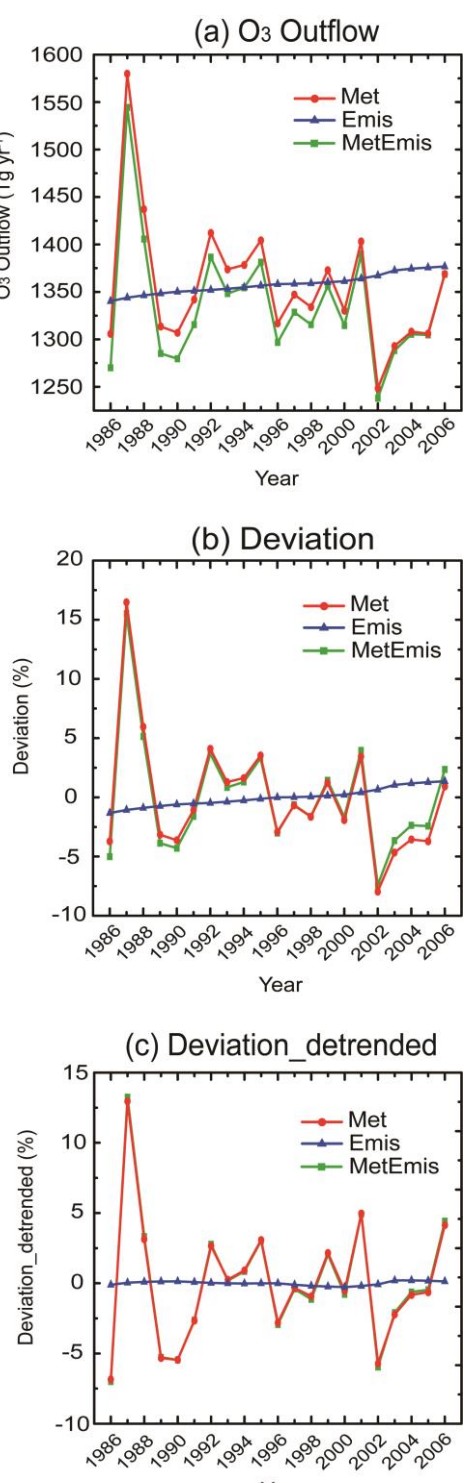

**Figure 7.** Evolution of (a) annual $O_3$ outflow fluxes (Tg yr$^{-1}$) across the meridional plane along 135 °E from 20 °N to 55 ° N, and from the surface to 100 hPa, over 1986–2006 in the Met, Emis, and MetEmis simulations; (b) the associated deviations from the mean (%); and (c) deviations from the mean (%) of the detrended $O_3$ outflow fluxes. The deviation from the mean (DEV) is defined in Sect. 4.2.

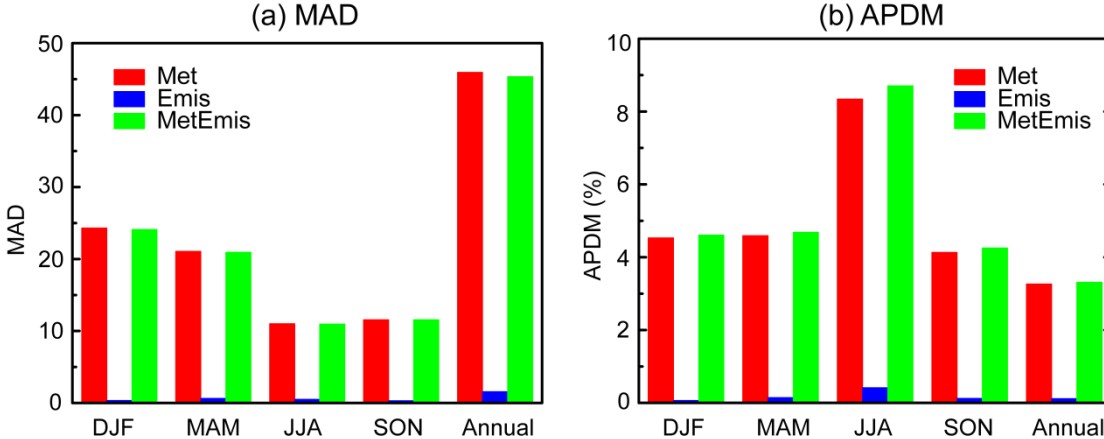

**Figure 8.** The MAD and APDM values of the detrended seasonal and annual $O_3$ outflow fluxes across the meridional plane along 135 °E from 20 °N to 55 °N, and from the surface to 100 hPa, over 1986–2006 in Met, Emis, and MetEmis simulations. Both the MAD and APDM are defined in Sect. 4.2. The units of MAD are Tg season$^{-1}$ for seasonal fluxes and Tg yr$^{-1}$ for annual fluxes.

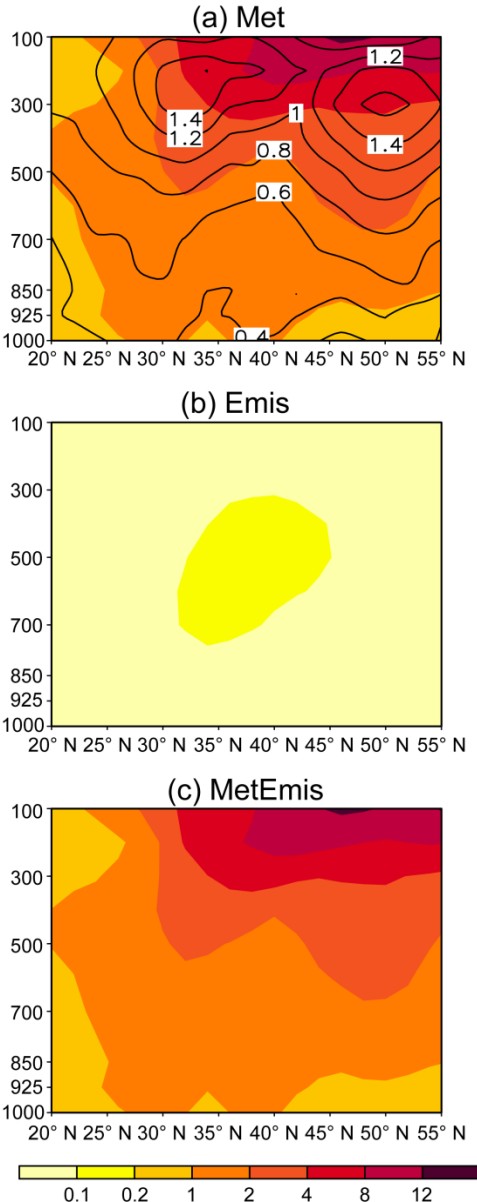

**Figure 9.** The pressure–latitude cross-sections along 135 °E of MAD values for detrended annual $O_3$ outflow fluxes and zonal winds over 1986–2006 in the Met, Emis, and MetEmis simulations. The MAD values for $O_3$ mass fluxes are shown by shades (units: kg yr$^{-1}$ m$^{-2}$), and the MAD values for winds are represented by contours (units: m s$^{-1}$).

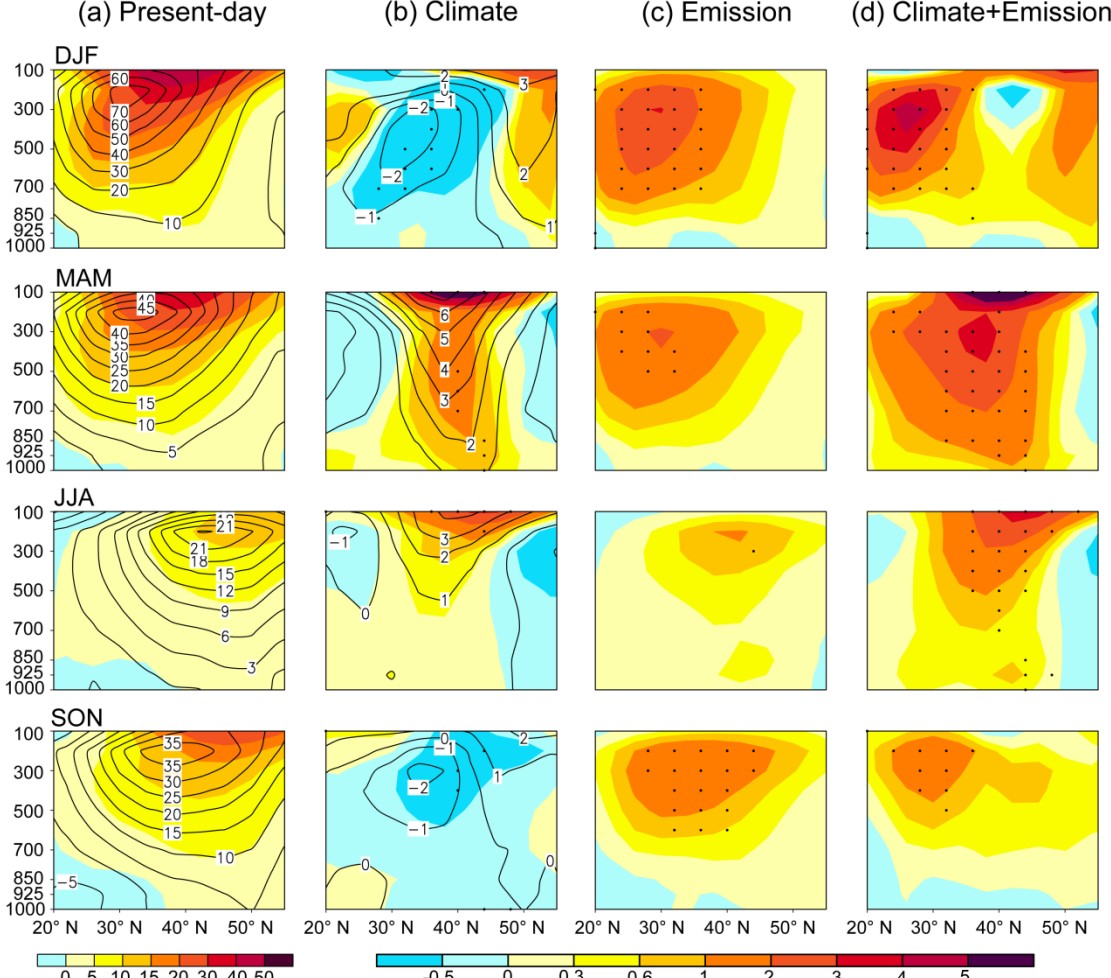

**Figure 10.** (a) The pressure–latitude cross-sections along $135\,°E$ of simulated present-day $O_3$ mass fluxes and zonal winds (Met2000Emis2000). Projected changes in $O_3$ mass fluxes and zonal winds from the present day (1996–2005) to the future (2046–2055) caused by (b) climate change alone (Met2050Emis2000 minus Met2000Emis2000); (c) changes in anthropogenic emissions alone (Met2000Emis2050 minus Met2000Emis2000); and (d) changes in both climate and anthropogenic emissions (Met2050Emis2050 minus Met2000Emis2000). The $O_3$ mass fluxes are shown by shades (units: kg season$^{-1}$ m$^{-2}$), and winds are represented by contours (units: m s$^{-1}$). The dotted areas are statistically significant at the 95 % level, as determined by a two-sample Student's $t$-test.