# Peer review of "Interannual variation, decadal trend, and future change in ozone outflow from East Asia"

_Atmospheric Chemistry and Physics, 2016_

## Referee Comment (RC1) · Anonymous Referee #1 · 8 Dec 2016

**Major comments**

This study uses a suite of chemical transport model experiments (GEOS-Chem) to examine the extent to which changes in anthropogenic emissions and meteorology influence the outflow of ozone from East Asia under present-day and future climate. The authors show that Asian NOx emissions almost doubled over the historical analysis period 1986-2006, along with increases in VOC emissions and global methane (Fig.1). However, their model with both emissions and meteorology varying over 1986-2006 shows little overall trend in the outflow of ozone from East Asia (Fig.6). This result contradicts with many prior studies suggesting that rising Asian emissions over the past 20-30 years contribute to raising baseline ozone downwind of Asia and over western North America. The referee believes that there are likely some fundamental flaws in the model experiments (or analysis approach). Further in-depth analyses are needed

to evaluate the modeled ozone response to emission trends.

The referee recommends the following analyses:

(1) Does the model (MetEmis) simulate significant increases in surface and free tropospheric ozone over East Asia during the period 1986-2006? How well do the modeled trends compare with observations? While long-term ozone observations over East Asia are very limited, there are some data available. Please see Section 3 and Figs 4-6 in the following manuscript and references therein:

Lin, M., Horowitz, L. W., Payton, R., Fiore, A. M., and Tonnesen, G.: US surface ozone trends and extremes from 1980–2014: Quantifying the roles of rising Asian emissions, domestic controls, wildfires, and climate, Atmos. Chem. Phys. Discuss., doi:10.5194/acp-2016-1093, in review, 2016, accessible at http://www.atmos-chem-phys-discuss.net/acp-2016-1093/

(2) This study defines the Asian ozone outflow as the ozone flux through the meridional plain along 135E from 20N-55N and from the surface to 100 hPa. If you restrict the calculation to the surface to 200-300 hPa or up to the tropopause, does the calculated O3 flux change substantially? I wonder if the O3 flux up to 100 hPa is overwhelmingly influenced by stratosphere-to-trposphere exchange (STE) and thus the emission-driven trend is swamped by interannual variability in STE.

(3) This study uses tropospherc column ozone (TCO) retrieved from TOMS/SBUV to evaluate their model simulation of TCO seasonal cycle and long-term trends (Figs 3 and 4). But how good are the TOMS TCO retrivals? TOMS TCO is possibly representative of mid- and upper tropospheric ozone variability. It is not expected to resolve ozone variability in the lower troposphere. So why use TOMS to evaluate the model?

(4) Fig.9 and associated discussions about the future changes.

Changes in atmospheric circulation on regional scales under future climate scenarios are known to have large uncertainty. The different models often yield different results

and large ensemble members are typically required.

Theodore G. Shepherd, Atmospheric circulation as a source of uncertainty in climate change projections, Nature Geoscience 7, 703–708 (2014), doi:10.1038/ngeo2253.

How many ensembles are included in your experiments? Rather than just showing your results, a through literature review is needed in Section 5 to place your results into context. What is the robust conclusion across the models in the published literature regarding changes in zonal winds and other circulation aspects over Asia under future climate? Do your model agree with the published work?

**Other minor comments**

**Page 2, Line 10**: Also cite Jacob et al., 1999 - the first paper on Asian influence on US ozone.

Jacob, D. J., Logan, J. A. Murti, P. P. Effect of rising Asian emissions on surface ozone in the United States. Geophys. Res. Lett. 26, 2175-2178 (1999).

**Page 3, Lines 9-10**: Also cite Lin et al. (2015b, GRL) - who found that measurement sampling biases substantially influence the ozone trends derived from sparse measurements over the western US originally reported by Cooper et al. (2010, Nature).

Lin, M., .W. Horowitz, O.R. Cooper, D. Tarasick, S. Conley, L.T. Iraci, B. Johnson, T. Leblanc, I. Petropavlovskikh, E.L. Yates (2015): Revisiting the evidence of increasing springtime ozone mixing ratios in the free troposphere over western North America, Geophysical Research Letter, 42, doi:10.1002/2015GL065311

**Page 3, Lines 13-15**: Also cite Schnell, J. L., M. J. Prather, B. Josse, V. Naik, L. W. Horowitz, G. Zeng, D. T. Shindell, and G. Faluvegi (2016), Effect of climate change on surface ozone over North America, Europe, and East Asia, Geophys. Res. Lett., 43, 3509-3518, doi: 10.1002/2016GL068060.

**Page 3, Lines 20-30**: Also explicitly discuss the results from Lin et al. (2014, Nature

Geoscience). They found that interannual variability of springtime Asian ozone outflow is strongly influenced by ENSO-related shifts in the subtropical jet stream. Transport of Asian pollution towards the eastern North Pacific during spring has weakened in the 2000s due to more frequent La Nina-like conditions.

---

## Author Comment (AC1) · 17 Dec 2016

**Response to the Major Comments of Referee #1**

We thank Reviewer #1 for the careful review of our work and would like to address your major concerns as follows. Our responses to your minor comments and the complete revision of our manuscript will be uploaded subsequently.

**Major comments**

This study uses a suite of chemical transport model experiments (GEOS-Chem) to examine the extent to which changes in anthropogenic emissions and meteorology influence the outflow of ozone from East Asia under present-day and future climate. The authors show that Asian NOx emissions almost doubled over the historical analysis period 1986-2006, along with increases in VOC emissions and global methane (Fig.1). However, their model with both emissions and meteorology varying over 1986-2006 shows little overall trend in the outflow of ozone from East Asia (Fig.6). This result contradicts with many prior studies suggesting that rising Asian emissions over the past 20-30 years contribute to raising baseline ozone downwind of Asia and over western North America. The referee believes that there are likely some fundamental flaws in the model experiments (or analysis approach). Further in-depth analyses are needed to evaluate the modeled ozone response to emission trends.

Response:

Thanks to the Reviewer for the careful review of our work. We would like to address your major concerns as follows:

1) In the manuscript, $O_3$ outflow from East Asia includes the effects of emissions in different regions of the world, other than Asian emissions alone, owing to the relatively long lifetime (~3 weeks) of $O_3$ (Fiore et al., 2002; Liao et al., 2006). Although the anthropogenic emissions of $O_3$ precursors in Asia increased a lot over 1986–2006, the global anthropogenic emissions of these precursors exhibited no significant trends over the past two decades (Figure 1(a) in the manuscript, similar to those reported by Lamarque et al., 2010), which explains in part the statistically insignificant decadal trend of $O_3$ outflow from East Asia.

2) The outflow flux of $O_3$ depends on both tropospheric $O_3$ concentrations and winds. The zonal winds exhibited large interannual variations over 1986–2006 (Figure 8(a) in the manuscript, also reported by Yuan and Ni, 2013; Du et al., 2016), which led to the large interannual variations in $O_3$ outflow flux. The emission-driven trend in $O_3$ flux is swamped by the large interannual variability in zonal winds. Therefore, with variations in both anthropogenic emissions and meteorological parameters, the simulated $O_3$ outflow fluxes showed statistically insignificant decadal trend.

3) Following the reviewer's suggestion (the following recommended analyses (1)), we have compared the trends in simulated $O_3$ concentrations with the observations. In fact, the simulated $O_3$ concentrations in our model exhibited statistically significant increasing trends over 1986–2006 (see our response to the following recommended analyses (1)), which verified the validity of the model experiments.

The referee recommends the following analyses:

(1) Does the model (MetEmis) simulate significant increases in surface and free

tropospheric ozone over East Asia during the period 1986–2006? How well do the modeled trends compare with observations? While long-term ozone observations over East Asia are very limited, there are some data available. Please see Section 3 and Figs 4-6 in the following manuscript and references therein:

Lin, M., Horowitz, L. W., Payton, R., Fiore, A. M., and Tonnesen, G.: US surface ozone trends and extremes from 1980–2014: Quantifying the roles of rising Asian emissions, domestic controls, wildfires, and climate, Atmos. Chem. Phys. Discuss., doi:10.5194/acp-2016-1093, in review, 2016, accessible at http://www.atmos-chemphys-discuss.net/acp-2016-1093/.

Response:

Following the reviewer's suggestion, we have calculated the trends in simulated (MetEmis) surface $O_3$ concentrations over East Asia, and compared with the observed trends collected from the above manuscript (Lin et al., 2016) and references therein. Figure R1 shows the comparison of simulated $O_3$ trends with observations. Although the model underestimates the observed trends, simulated $O_3$ concentrations at all stations exhibit statistically significant increasing trends. The modeled $O_3$ trends had low biases in previous studies (Tanimoto et al., 2009; Parrish et al., 2014); Parrish et al. (2014) compared $O_3$ trends simulated by three chemistry-climate models with observations at Asian sites, and reported that one model captured less than one third of the observed increasing trend, and the other two models suggested no significant increasing trends.

[Figure]

**Figure R1.** Comparison of modeled (MetEmis) trends in annual-mean $O_3$ concentrations with observations.

(2) This study defines the Asian ozone outflow as the ozone flux through the meridional plain along 135E from 20N-55N and from the surface to 100 hPa. If you restrict the calculation to the surface to 200-300 hPa or up to the tropopause, does the calculated $O_3$ flux change substantially? I wonder if the $O_3$ flux up to 100 hPa is overwhelmingly influenced by stratosphere-to-troposphere exchange (STE) and thus the emission-driven trend is swamped by interannual variability in STE.

Response:

1) As we describe in Section 2.2 of our manuscript, we do not consider the interannual variations in stratosphere–troposphere exchange (STE) of $O_3$ in this study. The model imposes a global annual mean cross-tropopause $O_3$ flux of 500

Tg yr$^{-1}$. We have added discussions on this issue in the text and in the Conclusion section of our revised manuscript.

2) Following the reviewer's suggestion, we have also calculated $O_3$ fluxes through the meridional plain along 135 °E from the surface to 200 hPa. Figure R2(a) and R2(b) show the evolutions of annual $O_3$ outflow fluxes across the meridional plane from the surface to 100 hPa and from the surface to 200 hPa, respectively. When we restrict the calculation to 200 hPa, the patterns of variations in $O_3$ fluxes are similar to those calculated from the surface to 100 hPa. With variations in both anthropogenic emissions and meteorological parameters (the MetEmis simulation), the simulated $O_3$ outflow shows large IAVs but a statistically insignificant ($P >$ 0.05) trend.

[Figure]

**Figure R2.** Evolution of annual $O_3$ outflow fluxes (Tg yr$^{-1}$) across the meridional plane along 135 ° E from 20 °N to 55 °N for (a) from the surface to 100 hPa and (b) from the surface to 200 hPa.

(3) This study uses tropospheric column ozone (TCO) retrieved from TOMS/SBUV to evaluate their model simulation of TCO seasonal cycle and long-term trends (Figs 3 and 4). But how good are the TOMS TCO retrievals? TOMS TCO is possibly representative of mid- and upper tropospheric ozone variability. It is not expected to resolve ozone variability in the lower troposphere. So why use TOMS to evaluate the model?

Response:

1) Tropospheric column ozone (TCO) can be retrieved from satellites TES, OMI/MLS, TOMS/SBUV, and so on. The TES and OMI are both on the EOS Aura satellite launched in July 2004. Therefore, TCO retrievals from TES or OMI/MLS are not available before year 2004 (the MetEmisB simulation in the manuscript is conducted over 1997–2006). However, TCO retrievals from TOMS/SBUV retrievals are available since 1979.

2) Fishman et al. (1996) compared tropospheric ozone fields derived from TOMS/SBUV with ozone measurements from aircraft, ozonesondes, and TOMS/SAGE. In general, TOMS/SBUV technique successfully captures the amount, large-scale gradients, and temporal variations of tropospheric column ozone. TOMS/SBUV technique has been extensively used to study the temporal-spatial distributions of TCO (Fishman and Balok, 1999; Fishman et al., 2003) and intercontinental transport of tropospheric ozone (Creilson et al., 2003).

(4) Fig.9 and associated discussions about the future changes. Changes in atmospheric circulation on regional scales under future climate scenarios are known to have large

uncertainty. The different models often yield different results and large ensemble members are typically required.

Theodore G. Shepherd, Atmospheric circulation as a source of uncertainty in climate change projections, Nature Geoscience 7, 703–708 (2014), doi:10.1038/ngeo2253.

How many ensembles are included in your experiments? Rather than just showing your results, a through literature review is needed in Section 5 to place your results into context. What is the robust conclusion across the models in the published literature regarding changes in zonal winds and other circulation aspects over Asia under future climate? Do your model agree with the published work?

Response:

1) The future simulation of $O_3$ outflow is driven by meteorological data from the Goddard Institute for Space Studies (GISS) general circulation model (GCM) 3. The GISS Model 3 is coupled with a "Q-flux" ocean as described in Wu et al. (2008). The GISS Model 3/GEOS-Chem combination has been used to project future aerosols and ozone over United States and China under future climate (Wu et al., 2008; Pye et al., 2009; Wang et al., 2013; Jiang et al., 2013).

2) Following the reviewer's suggestion, we have compared the future changes in zonal winds in our revised manuscript with those reported by previous studies: "The projected future changes in zonal winds are consistent with previous studies. By analyzing 18 CMIP5 models, Huang and Wang. (2016) assessed the future changes in atmospheric circulation during spring over East Asia. They found that although different models projected different changes (even in sign) in zonal winds, the ensemble mean of five better-skill models among the 18 CMIP5 models exhibited overall increases in zonal winds throughout the whole troposphere, which agrees with our simulation. Based on 31 (29)-model ensemble mean results, Jiang and Tian. (2013) showed that westerlies along 135 °E during winter (summer) were projected to weaken (strengthen) south of 40 °N. The projected patterns of future changes in westerlies during winter and summer are also captured by our model."

**References:**

Creilson, J. K., Fishman J., Wozniak, A. E.: Intercontinental transport of tropospheric ozone :a study of its seasonal variability across the North Atlantic utilizing tropospheric ozone residuals and its relationship to the North Atlantic Oscillation, Atmos. Chem. Phys., 3, 2053–2066, doi:10.5194/acp-3-2053-2003, 2003.

Du, Y., Li, T., Xie, Z., and Zhu, Z.: Interannual variability of the Asian subtropical westerly jet in boreal summer and associated with circulation and SST anomalies, Clim. Dyn., 46, 2673–2688, doi:10.1007/s00382-015-2723-x, 2016.

Fiore, A. M., Jacob, D. J., Bey, I., Yantosca, R. M., Field, B. D., Fusco, A. C., and Wilkinson, J. G.: Background ozone over the United States in summer: Origin, trend, and contribution to pollution episodes, J. Geophys. Res., 107, D15, doi:10.1029/2001JD000982, 2002.

Fishman J., and Balok A. E.: Calculation of daily tropospheric ozone residuals using TOMS and empirically improved SBUV measurements: Application to an ozone

pollution episode over the eastern United States, J. Geophys. Res., 104, 30319–30340, doi: 10.1029/1999JD900875, 1999.

Fishman, J., Brackett, V. G., Browell, E. V., and Grant, W. B.: Tropospheric ozone derived from TOMS/SBUV measurements during TRACE A, J. Geophys. Res., 101, 24069–24082, doi:10.1029/95JD03576, 1996.

Fishman, J., Wozniak, A. E., and Creilson, J. K.: Global distribution of tropospheric ozone from satellite measurements using the empirically corrected tropospheric ozone residual technique: Identification of the regional aspects of air pollution, Atmos. Chem. Phys., 3, 893–907, doi:10.5194/acp-3-893-2003, 2003.

Huang, W. R., and Wang, S. Y. S.: Future changes in propagating and non-propagating diurnal rainfall over East Asia, Clim. Dyn., 1–15, doi: 10.1007/s00382-016-3348-4, 2016.

Jiang D. B., and Tian, Z. P.: East Asian monsoon change for the 21st century: Results of CMIP3 and CMIP5 models, Chinese Sci. Bull., 58, 1427–1435, doi:10.1007/s11434-012-5533-0, 2013.

Jiang, H., Liao, H., Pye, H. O. T., Wu, S., Mickley, L. J., Seinfeld, J. H., and Zhang, X. Y.: Projected effect of 2000–2050 changes in climate and emissions on aerosol levels in China and associated transboundary transport, Atmos. Chem. Phys., 13, 7937–7960, doi:10.5194/acp-13-7937-2013, 2013.

Lamarque, J. F., et al., Historical (1850–2000) gridded anthropogenic and biomass burning emissions of reactive gases and aerosols: methodology and application, Atmos. Chem. Phys., 10, 7017–7039, doi:10.5194/acp-10-7017-2010, 2010.

Liao, H., Chen, W. T., and Seinfeld, J. H.: Role of climate change in global predictions of future tropospheric ozone and aerosols, J. Geophys. Res., 111, D12304, doi:10.1029/2005JD006852, 2006.

Parrish, D. D., et al., Long-term changes in lower tropospheric baseline ozone concentrations: Comparing chemistry-climate models and observations at northern midlatitudes, J. Geophys. Res., 119(9), 5719–5736, doi: 10.1002/2013JD021435, 2014.

Pye, H. O. T., Liao, H., Wu, S., Mickley, L. J., Jacob, D. J., Henze, D. K., and Seinfeld, J. H.: Effect of changes in climate and emissions on future sulfate-nitrate-ammonium aerosol levels in the United States, J. Geophys. Res., 114, D01205, doi:10.1029/2008JD010701, 2009.

Tanimoto, H., Ohara, T., and Uno, I.: Asian anthropogenic emissions and decadal trends in springtime tropospheric ozone over Japan: 1998–2007, Geophys. Res. Lett., 36, L23802, doi:10.1029/2009GL041382, 2009.

Wang, Y., Shen, L., Wu, S., Mickley, L., He, J., and Hao, J.: Sensitivity of surface ozone over China to 2000–2050 global changes of climate and emissions, Atmos. Environ., 75, 374–382, doi:10.1016/j.atmosenv.2013.04.045, 2013.

Wu, S. L., Mickley, L. J., Leibensperger, E. M., Jacob, D. J., Rind, D., and Streets, D. G.: Effects of 2000–2050 global change on ozone air quality in the United States, J. Geophys. Res.-Atmos., 113, D06302, doi:10.1029/2007jd008917, 2008.

Yuan, S., and Ni, J.: Interannual and interdecadal variations of the upper level zonal circulation, The Science Education Article Collects, 11, 100–104, 2013.

---

## Referee Comment (RC2) · Anonymous Referee #2 · 24 Dec 2016

A review of "Interannual variation, decadal trend, and future change in ozone outflow from East Asia" by Jia Zhu et al. submitted to ACP

General comments: The paper describes a model analysis of the past (1986-2006) and future (2000 vs. 2050) changes in the continental outflow of tropospheric ozone from East Asia. For the past and future changes, the authors ran the GEOS-Chem model driven by meteorological fields from GEOS4 and GISS GCM3 (under SRES A1B scenario), respectively. Basically the topics of the paper are of substantial interest. However, I found that the paper is rather descriptive and the discussion is not thorough. In many parts of the paper the authors show statistical results and interpretation rather than in-depth analyses that they could do with such a suite of model simulations. My another concern is that the authors' approach using the SREA A1B scenario now sounds old model sets, and I wonder why the authors did not try the

simulations with RCP scenarios. Well, reserving this criticism, the paper still needs to be more focused on a new science with respect to continental outflow of ozone from East Asia that the authors can deliver from the current model runs.

Major comments:

(1) Why SRES A1B scenario? - I think this scenario is now out of date and would not be realistic for the future, suggesting the model studies less useful than before, say several years ago. If the authors stick to the SRES scenario, they would need to justify why they used this scenario not the RCP one. Also, I found that the discussions read a bit superficial, with a lot of interpretations by referring to previously published papers based on similar model settings with the SRES A1B scenario (i.e., Wu et al., Pye et al., Jiang et al.). The authors should focus on a new science with respect to continental outflow of ozone from East Asia, provide in-depth analysis in terms of meteorological and climatic mechanisms or key factors. In Abstract, the authors mentioned "Sensitivity simulations indicated that the large IAVs of O3 outflow fluxes were mainly caused by the variations in meteorological conditions.", but this statement reads rather general. What meteorological factors or mechanisms are key for IAV? The authors showed statistical analysis but the mechanisms behind the large IAV is much more informative to the community.

(2) More robust model validation (Section 3) - Because of the large uncertainty in the retrieval of tropospheric ozone, comparison to satellite is not a robust way to quantitatively evaluate the model performance for the lower tropospheric ozone, in particular. The authors can make satellite comparisons with the reasons they mentioned in the Reply to the other reviewer, but why don't the authors evaluate the model by comparing to surface and sondes observations available in East Asia? I strongly believe that the model validation should be intensively made on seasonal basis since the authors are discussing the past and future ozone flux based on the model runs. The data from EANET are often used in evaluating the regional and global models by many groups in Asia (e.g., MICS-Asia) and in the international projects (e.g., HTAP) (e.g., Nagashima

et al., ACP, 2010; Li et al., 2008). In Figures 3 and 4, the model overestimated the satellite-derived TCO over central-eastern China through the western North Pacific, and the phase of the seasonal cycle in TCO is not as great as the current state-of-science models could be. I do not see the model doing a good job in reproducing the distributions and seasonal cycles, so cannot be positive to support the further analysis. The model overestimates TCO in spring, so this would give the overestimates in the calculated eastward flux. On the other hand, maybe the satellite-derived TCO is not too low (Figure 3), or the maximum shifts later than should be (Figure 4). I would encourage the authors to examine the model-observation comparison for the boundary layer, and middle and upper troposphere. Recent paper by Tanimoto, Zbinden, et al. (2015) showed robust observations for the seasonal cycles and interannual variations over Japan, and would be useful for this comparison.

(3) Is 135 degE appropriate? - The authors mentioned in the title "ozone outflow from East Asia" and used a longitudinal transect at 135 degE to diagnose the eastward flux of ozone. I wonder why at 135 degE, not 120 degE, to be more close to ozone production region in central-eastern China. I think, if the authors look at the flux at 120 degE, they would obtain higher signals in the ozone flux, and this would be much more direct in interpreting the model simulations. Also, the authors mainly discuss central-eastern China or North China Plain, rather than whole East Asia. This should be explicitly phrased, for example, "outflow from central-eastern China", since this paper is not looking at the impacts on the western North America but focusing on export region.

Specific comments:

Abstract, L22: insignificant decadal trend of -2.2%/decade. Add +/- uncertainty, or just delete the number here.

L28-29: spring and summer. The maritime flow from the Pacific Ocean is predominant in summer. Is summer really effective in the enhancement of continental outflow? I do

not see strong enhancement in summer in Figure 9.

L31: important implications for long-term air quality planning. For whom? For US? For northern midlatitudes? For China? For East Asia?

P2, L8-11: ... influences ozone air quality in the downwind regions, such as the US and Canada. Downwind regions are not only the western US, but should include the neighboring regions and the Pacific Ocean. Ou-Yang et al. paper is already cited here, so the sentence should be rephrased to be something like "... such as the western North Pacific through the western North America", and add some other references, for example, papers reporting long-range transport to Korea, Japan, and the Northern Pacific (Han et al., ACP, 2015; Tanimoto et al., GRL, 2005; Pochanart et al., 2015, and many others!).

P3, L8-12: Tanimoto, AE, 2009 should be cited here (decadal trends of ...)

Figure 4: The authors showed comparison of TCO for GEOS-Chem and TOMS/SBUV, suggesting the biases in the model. Again, why don't the authors make comparison to the surface and sonde observations?

Also, in P8, L10-13, the authors state that "although GEOS-Chem overestimates TCO values over eastern China and the western Pacific Ocean, the model exhibits reasonable performance in simulating the spatiotemporal distributions of the tropospheric ozone column burden over China and downwind regions, which lends us confidence to simulate the temporal evolutions of the Asian ozone outflow." I would not agree with this statement, since model and satellite are quite different in the tail of outflow from China (the region of >40 DU, shown in orange), and this difference would lead to large biases in calculating outflow flux in particular, as the authors set the diagnosis line at 135 degE, off China, and over Japan. Also, technically, the authors said "the western Pacific Ocean" here and also in the Figure 4 caption, but the region where the authors pointed is mostly Japan, so the description must be accurately modified.

P9, Section 4.2 IAV and decadal trends: The authors basically said that the influence of Met. is larger than Emiss., which makes sense if they diagnosed at 135 degE, off the Asian continent, where Asian monsoon impacts are substantial.

A number of important references are missing: Pochanart, P. et al, 2015, Boundary Layer Ozone Transport from Eastern China to Southern Japan: Pollution Episodes Observed during Monsoon Onset in 2004, Asian J. Atmos. Environ. 9, 48-56.

Tanimoto, H., et al., 2005, Significant latitudinal gradient in the surface ozone spring maximum over East Asia, Geophys. Res. Lett. 32, L21805.

Tanimoto, H., 2009, Increase in springtime tropospheric ozone at a mountainous site in Japan for the period 1998–2006, Atmos. Environ., 43, 1358-1363.

Nagashima, T., et al., 2010, The relative importance of various source regions on East Asian surface ozone, Atmos. Chem. Phys., 10, 11305-11322, doi:10.5194/acp-10-11305-20109077-9120.

Li, J., et al., 2008, Regional-scale modeling of near-ground ozone in the Central East China, source attributions and an assessment of outflow to East Asia - The role of regional-scale transport during MTX2006, Atmos. Chem. Phys., 8, doi: 10.5194/acpd-8-13159-2008.

Tanimoto, H., et al., 2015, Consistency of tropospheric ozone observations made by different platforms and techniques in the global databases, Tellus B, 67, 27073, doi: 10.3402/tellusb.v67.27073.
* * *

---

## Author Comment (AC2) · 22 Jan 2017

**Response to Referee #2**

We would like to thank Referee #2 for the careful review of our work and address your concerns as follows. The reviewer's comments are shown with black font and our replies including the updates to the manuscript are highlighted in blue below.

**General comments**: The paper describes a model analysis of the past (1986-2006) and future (2000 vs. 2050) changes in the continental outflow of tropospheric ozone from East Asia. For the past and future changes, the authors ran the GEOS-Chem model driven by meteorological fields from GEOS4 and GISS GCM3 (under SRES A1B scenario), respectively. Basically the topics of the paper are of substantial interest. However, I found that the paper is rather descriptive and the discussion is not thorough. In many parts of the paper the authors show statistical results and interpretation rather than in-depth analyses that they could do with such a suite of model simulations. My another concern is that the authors' approach using the SREA A1B scenario now sounds old model sets, and I wonder why the authors did not try the simulations with RCP scenarios. Well, reserving this criticism, the paper still needs to be more focused on a new science with respect to continental outflow of ozone from East Asia that the authors can deliver from the current model runs.

**Response:**

1) We have added more detailed discussions and analyses in the revised manuscript, including:
   - Analyses on the key factor that drove the large interannual variations in continental $O_3$ outflow. See our response to your major comment (1);
   - More comprehensive validation on simulated surface-layer $O_3$ concentrations by using $O_3$ measurements from WDCGG and EANET, and the validation on simulated $O_3$ concentrations for the boundary layer, middle and upper troposphere by using the ozonesonde data from WOUDC. See our response to your major comment (2);
   - Comparisons with published literature regarding future changes in zonal winds. See our response to the recommended analyses (4) of Reviewer #1;
   - "Uncertainty Discussion" section (Section 6) to discuss the uncertainties associated with our model results.
2) The RCP scenarios have been compared with the SRES scenarios in previous studies (Lamarque et al., 2011; Riahi et al., 2011; van Vuuren et al., 2011; Fiore et al., 2012). For future air pollutant emissions, the RCPs assume uniformly an aggressive reduction, whereas the SRES scenarios allow unconstrained growth (Fiore et al., 2012). These two sets of projections likely bracket possible futures (Fiore et al., 2012). Therefore, the SRES A1B scenario is still used in recent studies to project future climate and $O_3$ (Lee et al., 2015; Redmond et al., 2015; Glotfelty et al., 2016; Sanderson and Ford, 2016).

**Major comments**:

(1) Why SRES A1B scenario? - I think this scenario is now out of date and would not be realistic for the future, suggesting the model studies less useful than before, say several years ago. If the authors stick to the SRES scenario, they would need to justify why they used this scenario not the RCP one. Also, I found that the discussions read a bit superficial, with a lot of interpretations by referring to previously published papers based on similar model settings with the SRES A1B scenario (i.e., Wu et al., Pye et al., Jiang et al.). The authors should focus on a new science with respect to continental outflow of ozone from East Asia, provide in-depth analysis in terms of meteorological and climatic mechanisms or key factors. In Abstract, the authors mentioned "Sensitivity simulations indicated that the large IAVs of O3 outflow fluxes were mainly caused by the variations in meteorological conditions.", but this statement reads rather general. What meteorological factors or mechanisms are key for IAV? The authors showed statistical analysis but the mechanisms behind the large IAV is much more informative to the community.

**Response:**

1) We have justified the use of SRES A1B scenario in our response to your general comments.

2) Following the Reviewer's suggestion, we have added the following discussions on the key factor for the large IAVs in the revised manuscript (the last paragraph of Section 4.2): "Variations in meteorological conditions can influence the IAVs of the $O_3$ outflow fluxes by changing $O_3$ concentrations over East Asia (Yang et al., 2014; Lou et al., 2015), and by altering zonal winds (Kurokawa et al., 2009). The $O_3$ outflow flux is simulated to correlate positively with zonal wind averaged over 20°–55° N along 135° E, with a high correlation coefficient of +0.71 for annual fluxes and zonal winds. The correlation coefficient between $O_3$ fluxes and zonal winds is calculated to be +0.96 during summer when the APDM values of $O_3$ outflow fluxes are maximum. The high correlation coefficients indicate that the variation in zonal winds is the key factor that leads to the large IAVs of $O_3$ outflow fluxes."

(2) More robust model validation (Section 3) - Because of the large uncertainty in the retrieval of tropospheric ozone, comparison to satellite is not a robust way to quantitatively evaluate the model performance for the lower tropospheric ozone, in particular. The authors can make satellite comparisons with the reasons they mentioned in the Reply to the other reviewer, but why don't the authors evaluate the model by comparing to surface and sondes observations available in East Asia? I strongly believe that the model validation should be intensively made on seasonal basis since the authors are discussing the past and future ozone flux based on the model runs. The data from EANET are often used in evaluating the regional and global models by many groups in Asia (e.g., MICS-Asia) and in the international projects (e.g., HTAP) (e.g., Nagashima et al., ACP, 2010; Li et al., 2008). In Figures 3 and 4, the model overestimated the satellite-derived TCO over central-eastern China through the western North Pacific, and the phase of the seasonal cycle in TCO is not as great as the current state-of science models could be. I do not see the model doing a good job in reproducing the distributions and seasonal cycles, so cannot be positive to support the further analysis. The model overestimates TCO in spring, so this would give the overestimates in the calculated eastward flux. On the other hand, maybe the satellite-derived TCO is not too low (Figure 3), or the maximum shifts later than should be (Figure 4). I would encourage the authors to examine the model-observation comparison for the boundary layer, and middle and upper troposphere. Recent paper by Tanimoto, Zbinden, et al. (2015) showed robust observations for the seasonal cycles and interannual variations over Japan, and would be useful for this comparison.

**Response:**

Following the Reviewer's suggestion, we have changed the comparisons with satellite data to the comparisons with surface and sondes observations (added new Table 4, Figure 3, and Figure 4) in the revised manuscript. The data from WDCGG and EANET are used to evaluate the simulated surface-layer $O_3$ concentrations, and the ozonesonde data from WOUDC are used to evaluate the simulated $O_3$ concentrations for the boundary layer, middle and upper troposphere. We have added the following descriptions on the comparisons in the second and the third paragraphs of Section 3: "Here, we conduct comparisons with measurements to evaluate whether the version of the GEOS-Chem model used in this study can capture the temporal variations of tropospheric $O_3$. We use observations of tropospheric $O_3$ available in East Asia as summarized in Table 4. Observations at two sites (Minamitorishima and Yonagunijima) are from the World Data Centre for Greenhouse Gases (WDCGG, www.ds.data.jma.go.jp/gmd/wdcgg/), and those at another two sites (Rishiri and Ogasawara) are from the Acid Deposition Monitoring Network in East Asia (EANET, www.eanet.asia/product/index.html), which are used to evaluate the simulated surface-layer $O_3$ concentrations. The four Japanese sites are "remote" sites in the downwind regions of China. Figure 3 compares the time series of monthly surface-layer $O_3$ mixing ratios simulated by MetEmisB with those measured by WDCGG and EANET. Simulated surface-layer $O_3$ levels agree well with observations at all the four stations. The model captures fairly well the seasonal cycles and interannual variations of surface $O_3$, with high correlation coefficients of 0.82–0.93 (Table 4). Generally, the GEOS-Chem model can capture the high values during early spring or winter when Asian $O_3$ outflow flux is the highest, but overestimates the low values during summer when Asian $O_3$ outflow is the minimum.

To evaluate the simulated $O_3$ concentrations for the boundary layer, middle and upper troposphere,

we use the ozonesonde data at two Japanese sites from World Ozone and Ultraviolet Radiation Data Centre (WOUDC, www.woudc.org). The information for the two sites (Naha and Tsukuba) is listed in Table 4. Figure 4 compares the time series of monthly $O_3$ mixing ratios simulated by MetEmisB with those measured by ozonesonde. Comparisons are shown for four altitudes in the troposphere. The GEOS-Chem model captures the seasonal cycles and interannual variations of tropospheric $O_3$ at all altitudes, with correlation coefficients ranging from 0.68 to 0.88 for Naha site, and from 0.55 to 0.76 for Tsukuba site. However, the agreement with ozonesonde in the lowermost layer (1000–850 hPa) seems to be poorer than that with WDCGG or EANET. It is noted that, the ground-based measurements (WDCGG or EANET) and simulation results are calculated from continuous data, while the ozonesondes are regularly launched at a fixed local time with a typical frequency of 1–2 weeks (Tanimoto et al., 2015). The inconsistency in sampling time may be responsible for the poorer agreement with ozonesonde."

**Table 4.** Information for the sites with $O_3$ measurements used in model evaluation.

| Site | Location | Database | Height | R[a] | NMB[b] (%) |
|---|---|---|---|---|---|
| Minamitorishima | 24.3 N, 154.0 E | WDCGG | surface | 0.92 | +12.7 |
| Yonagunijima | 24.5 N, 123.0 E | WDCGG | surface | 0.93 | +12.6 |
| Rishiri | 45.1 N, 141.2 E | EANET | surface | 0.82 | +2.4 |
| Ogasawara | 27.1 N, 142.2 E | EANET | surface | 0.90 | +29.6 |
| Naha | 26.2 N, 127.7 E | WOUDC | 500–300 hPa | 0.68 | –2.61 |
| | | | 700–500 hPa | 0.77 | +16.4 |
| | | | 850–700 hPa | 0.85 | +24.3 |
| | | | 1000–850 hPa | 0.88 | +39.5 |
| Tsukuba | 36.1 N, 140.1 E | WOUDC | 500–300 hPa | 0.55 | +15.8 |
| | | | 700–500 hPa | 0.76 | +12.3 |
| | | | 850–700 hPa | 0.76 | +8.61 |
| | | | 1000–850 hPa | 0.60 | +8.5 |

[a] Correlation coefficient (R) between the observed and simulated monthly $O_3$ mixing ratios.

[b] Normalized mean bias (NMB, %) between the observed and simulated monthly $O_3$ mixing ratios.

[Figure]

**Figure 3.** Time series of monthly surface-layer O$_3$ mixing ratios measured by WDCGG and EANET (blue line), and simulated by MetEmisB (red line). (a) Minamitorishima and (b) Yonagunijima are WDCGG sites, and (c) Rishiri and (d) Ogasawara sites are EANET sites.

[Figure]

**Figure 4.** Time series of monthly $O_3$ mixing ratios measured by ozonesonde (blue line), and simulated by MetEmisB (red line). (a) Naha and (b) Tsukuba are ozonesonde sites from WOUDC. Comparisons are shown for four altitude levels in the troposphere.

(3) Is 135 degE appropriate? - The authors mentioned in the title "ozone outflow from East Asia" and used a longitudinal transect at 135 degE to diagnose the eastward flux of ozone. I wonder why at 135 degE, not 120 degE, to be more close to ozone production region in central-eastern China. I think, if the authors look at the flux at 120 degE, they would obtain higher signals in the ozone flux, and this would be much more direct in interpreting the model simulations. Also, the authors mainly discuss central-eastern China or North China Plain, rather than whole East Asia. This should be explicitly phrased, for example, "outflow from central-eastern China", since this paper is not looking at the impacts on the western North America but focusing on export region.

**Response:**

We calculate $O_3$ flux through the vertical plane along $135\,°$E, because $135\,°$E is the easternmost boundary of China (i.e., Wusuli River in Northeastern China). Following the Reviewer's suggestion, we have also calculated the $O_3$ outflow flux along $120\,°$E. Figure H1 shows the evolutions of annual $O_3$

outflow fluxes across the meridional plane along (a) 135 °E and (b) 120 °E over 1986–2006 in the Met, Emis, and MetEmis simulations. The variations in $O_3$ fluxes calculated at 120 °E are similar to those calculated at 135 °E. Both figures show that, with variations in both anthropogenic emissions and meteorological parameters (the MetEmis simulation), the simulated $O_3$ outflow shows large IAVs but a statistically insignificant ($P > 0.05$) trend. We have added the above discussion in the Uncertainty Discussion section (Section 6) of our revised manuscript.

Because the variations in $O_3$ fluxes calculated at 135 °E are similar to those calculated at 120 °E, we retain the calculations along 135 °E and the description of "ozone outflow from East Asia" in the revised manuscript.

[Figure]

**Figure H1.** Evolution of annual $O_3$ outflow fluxes (Tg yr$^{-1}$) across the meridional plane along (a) 135 ° E, and (b) 120 °E, from 20 °N to 55 °N and from the surface to 100 hPa over 1986–2006 in the Met, Emis, and MetEmis simulations.

**Specific comments**:

Abstract, L22: insignificant decadal trend of -2.2%/decade. Add +/- uncertainty, or just delete the number here.
**Response:**
    We have deleted the number in our revised manuscript.

L28-29: spring and summer. The maritime flow from the Pacific Ocean is predominant in summer. Is summer really effective in the enhancement of continental outflow? I do not see strong enhancement in summer in Figure 9.
**Response:**
    Although the absolute value of the increase in $O_3$ outflow during summer is not large, the percentage increase in $O_3$ outflow during summer is 14.5% (Table 5 in the revised manuscript). The large percentage increase can be mainly attributed to the enhancement in zonal winds. Based on 29-model ensemble mean results, Jiang and Tian (2013) also showed that the westerlies along 135 °E during summer would strengthen in future climate.

L31: important implications for long-term air quality planning. For whom? For US? For northern midlatitudes? For China? For East Asia?
**Response:**
    We have revised the sentence as "have important implications for long-term air quality planning for the downwind regions of China, such as Japan and US" in the Abstract of our revised manuscript.

P2, L8-11: . . . influences ozone air quality in the downwind regions, such as the US and Canada. Downwind regions are not only the western US, but should include the neighboring regions and the Pacific Ocean. Ou-Yang et al. paper is already cited here, so the sentence should be rephrased to be something like ". . . such as the western North Pacific through the western North America", and add some other references, for example, papers reporting long-range transport to Korea, Japan, and the

Northern Pacific (Han et al., ACP, 2015; Tanimoto et al., GRL, 2005; Pochanart et al., 2015, and many others!).

**Response:**

    We have replaced it by "such as the western North Pacific through the western North America", and added the following references: "Tanimoto et al., 2005; Kim et al., 2006; Li et al., 2008; Kurokawa et al., 2009; Nagashima et al., 2010; Han et al., 2015; Pochanart et al., 2015" (the second paragraph of Section 1).

P3, L8-12: Tanimoto, AE, 2009 should be cited here (decadal trends of . . .)

**Response:**

    We have cited the reference (Tanimoto, 2009) in the revised manuscript (the fourth paragraph of Section 1).

Figure 4: The authors showed comparison of TCO for GEOS-Chem and TOMS/SBUV, suggesting the biases in the model. Again, why don't the authors make comparison to the surface and sonde observations?

**Response:**

    As suggested by the Reviewer, we have conducted comparisons with surface-layer $O_3$ measurements from WDCGG and EANET, and with ozonesonde measurements from WOUDC for the boundary layer, middle and upper troposphere $O_3$ in the revised manuscript. See our response to your major comment (2).

Also, in P8, L10-13, the authors state that "although GEOS-Chem overestimates TCO values over eastern China and the western Pacific Ocean, the model exhibits reasonable performance in simulating the spatiotemporal distributions of the tropospheric ozone column burden over China and downwind regions, which lends us confidence to simulate the temporal evolutions of the Asian ozone outflow." I would not agree with this statement, since model and satellite are quite different in the tail of outflow from China (the region of >40 DU, shown in orange), and this difference would lead to large biases in calculating outflow flux in particular, as the authors set the diagnosis line at 135 degE, off China, and over Japan. Also, technically, the authors said "the western Pacific Ocean" here and also in the Figure 4 caption, but the region where the authors pointed is mostly Japan, so the description must be accurately modified.

**Response:**

    As the Reviewer pointed out in major comment (2), because of the large uncertainty in the retrieval of tropospheric ozone, comparison to satellite is not a robust way to quantitatively evaluate the model performance. Therefore, we have changed the comparisons with satellite to the comparisons with surface and sondes observations in the revised manuscript. See our response to your major comment (2). The old Figure 4 and associated description have been deleted and replaced by the comparisons with surface and sondes observations.

P9, Section 4.2 IAV and decadal trends: The authors basically said that the influence of Met. is larger than Emiss., which makes sense if they diagnosed at 135 degE, off the Asian continent, where Asian monsoon impacts are substantial.

**Response:**

    Following your suggestions in major comment (3), we have calculated the $O_3$ outflow flux along $120\,°\,E$, more close to the Asian continent. It is concluded from Figure H1 that the variations in $O_3$ fluxes calculated at $120\,°\,E$ are similar to those calculated at $135\,°\,E$. With variations in both anthropogenic emissions and meteorological parameters (the MetEmis simulation), the simulated $O_3$ outflow along $120\,°\,E$ also shows large IAVs but a statistically insignificant ($P > 0.05$) trend. The two curves from the Met and MetEmis simulations almost coincide with each other, indicating the dominant role of variations in meteorological parameters in the IAVs of the Asian $O_3$ outflow flux.

A number of important references are missing:

Pochanart, P. et al, 2015, Boundary Layer Ozone Transport from Eastern China to Southern Japan: Pollution Episodes Observed during Monsoon Onset in 2004, Asian J. Atmos. Environ. 9, 48-56.

Tanimoto, H., et al., 2005, Significant latitudinal gradient in the surface ozone spring maximum over East Asia, Geophys. Res. Lett. 32, L21805.

Tanimoto, H., 2009, Increase in springtime tropospheric ozone at a mountainous site in Japan for the period 1998–2006, Atmos. Environ., 43, 1358-1363.

Nagashima, T., et al., 2010, The relative importance of various source regions on East Asian surface ozone, Atmos. Chem. Phys., 10, 11305-11322, doi:10.5194/acp-10-11305-20109077-9120.

Li, J., et al., 2008, Regional-scale modeling of near-ground ozone in the Central East China, source attributions and an assessment of outflow to East Asia - The role of regional-scale transport during MTX2006, Atmos. Chem. Phys., 8, doi: 10.5194/acpd-8-13159-2008.

Tanimoto, H., et al., 2015, Consistency of tropospheric ozone observations made by different platforms and techniques in the global databases, Tellus B, 67, 27073, doi:10.3402/tellusb.v67.27073.

**Response:**

  We have added the above references in proper places of the revised manuscript.

**References:**

Fiore, A. M., et al.: Global air quality and climate, Chem. Soc. Rev., 41, 6663–6683, doi:10.1039/c2cs35095e, 2012.

Glotfelty, T., Zhang, Y., Karamchandani, P., and Streets, D. G.: Changes in future air quality, deposition, and aerosol-cloud interactions under future climate and emission scenarios, Atmos. Environ., 139, 176–191, doi:10.1016/j.atmosenv.2016.05.008, 2016.

Han, J., Shin, B., Lee, M., Hwang, G., Kim, J., Shim, J., Lee, G., and Shim, C.: Variations of surface ozone at Ieodo Ocean Research Station in the East China Sea and the influence of Asian outflows, Atmos. Chem. Phys., 15, 12611–12621, doi:10.5194/acp-15-12611-2015, 2015.

Jiang, D. B., and Tian, Z. P.: East Asian monsoon change for the 21st century: Results of CMIP3 and CMIP5 models, Chinese Sci. Bull., 58, 1427–1435, doi:10.1007/s11434-012-5533-0, 2013.

Kim, J. H., Lee, H. J., and Lee, S. H.: The characteristics of tropospheric ozone seasonality observed from ozone soundings at Pohang, Korea, Environ. Monit. Assess., 118, 1–12, doi:10.1007/s10661-006-0772-7, 2006.

Kurokawa, J., Ohara, T., Uno, I., Hayasaki, M., and Tanimoto, H.: Influence of meteorological variability on interannual variations of springtime boundary layer ozone over Japan during 1981–2005, Atmos. Chem. Phys., 9, 6287–6304, doi:10.5194/acp-9-6287-2009, 2009.

Lamarque, J. F., Kyle, G. P., Meinshausen, M., Riahi, K., Smith, S. J., van Vuuren, D. P., Conley, A. J., and Vitt, F.: Global and regional evolution of short-lived radiatively-active gases and aerosols in the Representative Concentration Pathways, Clim. Change, 109, 191–212, doi:10.1007/s10584-011-0155-0, 2011.

Lee, J. B., Cha, J. S., Hong, S. C., Choi, J. Y., Myoung, J. S., Park, R. J., Woo, J. H., Ho, C., Han, J. S., and Song, C. K.: Projections of summertime ozone concentration over East Asia under multiple IPCC SRES emission scenarios, Atmos. Environ., 106, 335–346, doi:10.1016/j.atmosenv.2015.02.019, 2015.

Li, J., Wang, Z., Akimoto, H., Yamaji, K., Takigawa, M., Pochanart, P., Liu, Y., Tanimoto, H., and Kanaya, Y.: Near-ground ozone source attributions and outflow in central eastern China during MTX2006, Atmos. Chem. Phys., 8, 7335–7351, doi:10.5194/acp-8-7335-2008, 2008.

Lou, S., Liao, H., Yang, Y., and Mu, Q.: Simulation of the interannual variations of tropospheric ozone over China: Roles of variations in meteorological parameters and anthropogenic emissions, Atmos. Environ., 122, 839–851, doi:10.1016/j.atmosenv.2015.08.081, 2015.

Nagashima, T., Ohara, T., Sudo, K., and Akimoto, H.: The relative importance of various source regions

on East Asian surface ozone, Atmos. Chem. Phys., 10, 11305–11322, doi:10.5194/acp-10-11305-2010, 2010.

Pochanart, P., Wang, Z., and Akimoto, H.: Boundary Layer Ozone Transport from Eastern China to Southern Japan: Pollution Episodes Observed during Monsoon Onset in 2004, Asian J. Atmos. Environ., 9, 48–56, doi:10.5572/ajae.2015.9.1.048, 2015.

Redmond, G., Hodges, K. I., Mcsweeney, C., Jones, R., and Hein, D.: Projected changes in tropical cyclones over Vietnam and the South China Sea using a 25 km regional climate model perturbed physics ensemble, Clim. Dyn., 45, 1983–2000, doi:10.1007/s00382-014-2450-8, 2015.

Riahi, K., Rao, S., Krey, V., Cho, C., Chirkov, V., Fischer, G., Kindermann, G., Nakicenovic, N., and Rafaj, P.: RCP 8.5—A scenario of comparatively high greenhouse gas emissions, Clim. Change, 109, 33–57, doi:10.1007/s10584-011-0149-y, 2011.

Sanderson, M. G., and Ford, G. P.: Projections of severe heat waves in the United Kingdom, Clim. Res., 71, 63–73, doi:10.3354/cr01428, 2016.

Tanimoto, H.: Increase in springtime tropospheric ozone at a mountainous site in Japan for the period 1998–2006, Atmos. Environ., 43, 1358–1363, doi:10.1016/j.atmosenv.2008.12.006, 2009.

Tanimoto, H., Sawa, Y., Matsueda, H., Uno, I., Ohara, T., Yamaji, K., Kurokawa, J., and Yonemura, S.: Significant latitudinal gradient in the surface ozone spring maximum over East Asia. Geophys. Res. Lett., 32, L21805, doi:10.1029/2005GL023514, 2005.

Tanimoto, H., Zbinden, R. M., Thouret, V., and Nédélec, P.: Consistency of tropospheric ozone observations made by different platforms and techniques in the global databases, Tellus B, 67, 27073, doi:10.3402/tellusb.v67.27073, 2015.

van Vuuren, D. P., et al.: The representative concentration pathways: an overview, Clim. Change, 109, 5–31, doi:10.1007/s10584-011-0148-z, 2011.

Yang, Y., Liao, H., and Li, J.: Impacts of the East Asian summer monsoon on interannual variations of summertime surface-layer ozone concentrations over China, Atmos. Chem. Phys., 14, 6867–6880, doi:10.5194/acp-14-6867-2014, 2014.

---

## Author Response (AR1)

**Responses to reviewers' comments on "Interannual variation, decadal trend, and future change in ozone outflow from East Asia" by J. Zhu et al. (MS No.: acp-2016-938)**

We would like to thank the reviewers for their comments that are helpful for improving the quality of our work. The manuscript has been revised accordingly, and our point-by-point responses are provided below. The reviewers' comments are shown with black font and our replies including the updates to the manuscript are highlighted in blue below. A marked-up manuscript version is also showing the changes made.

**Response to Reviewer #1**

**Major comments**

This study uses a suite of chemical transport model experiments (GEOS-Chem) to examine the extent to which changes in anthropogenic emissions and meteorology influence the outflow of ozone from East Asia under present-day and future climate. The authors show that Asian NOx emissions almost doubled over the historical analysis period 1986-2006, along with increases in VOC emissions and global methane (Fig.1). However, their model with both emissions and meteorology varying over 1986-2006 shows little overall trend in the outflow of ozone from East Asia (Fig.6). This result contradicts with many prior studies suggesting that rising Asian emissions over the past 20-30 years contribute to raising baseline ozone downwind of Asia and over western North America. The referee believes that there are likely some fundamental flaws in the model experiments (or analysis approach). Further in-depth analyses are needed to evaluate the modeled ozone response to emission trends.

**Response:**

We would like to address your major concerns as follows:

1) In the manuscript, $O_3$ outflow from East Asia includes the effects of emissions in different regions of the world, other than Asian emissions alone, owing to the relatively long lifetime (~3 weeks) of $O_3$ (Fiore et al., 2002; Liao et al., 2006). Although the anthropogenic emissions of $O_3$ precursors in Asia increased a lot over 1986–2006, the global anthropogenic emissions of these precursors ($NO_x$, CO, and NMVOCs) exhibited no significant trends over the past two decades (see Figure 1(a) in the revised manuscript, the trends we present are similar to those reported by Lamarque et al. (2010)), which explains in part the statistically insignificant decadal trend of $O_3$ outflow from East Asia.

2) The outflow flux of $O_3$ depends on both tropospheric $O_3$ concentrations and winds. As pointed out by the Reviewer, $O_3$ concentrations in East Asia exhibited an increasing trend in the past two decades (Ding et al., 2008a; Wang et al., 2009b; Xu et al., 2016). However, the zonal winds exhibited large interannual variations over 1986–2006 (Figure 9(a) in the revised manuscript, also reported by Yuan and Ni (2013) and Du et al. (2016)), which led to the large interannual variations in $O_3$ outflow flux. The emission-driven trend in $O_3$ flux was swamped by the large interannual variability in zonal winds. Therefore, with variations in both anthropogenic emissions and meteorological parameters, the simulated $O_3$ outflow fluxes showed statistically insignificant decadal trend.

3) Following the Reviewer's suggestion (the recommended analysis (1) below), we have compared the trends in simulated $O_3$ concentrations with observations. In fact, the simulated $O_3$ concentrations in our model exhibited statistically significant increasing trends over 1986–2006 (see our responses to the recommended analyses below), which verified the validity of our model experiments.

**The referee recommends the following analyses:**

(1) Does the model (MetEmis) simulate significant increases in surface and free tropospheric ozone over East Asia during the period 1986–2006? How well do the modeled trends compare with observations? While long-term ozone observations over East Asia are very limited, there are some data available. Please see Section 3 and Figs 4-6 in the following manuscript and references therein:

Lin, M., Horowitz, L. W., Payton, R., Fiore, A. M., and Tonnesen, G.: US surface ozone trends and extremes from 1980–2014: Quantifying the roles of rising Asian emissions, domestic controls, wildfires, and climate, Atmos. Chem. Phys. Discuss., doi:10.5194/acp-2016-1093, in review, 2016, accessible at

**Response:**

Following the Reviewer's suggestion, we have added new Figure 5 and the following descriptions on the comparison in the fourth paragraph of Section 3 of our revised manuscript: "The increasing trend in surface-layer $O_3$ in East Asia over the past two decades was reported by previous studies (Ding et al., 2008a; Wang et al., 2009b; Xu et al., 2016). Figure 5 compares the simulated trends in annual-mean surface-layer $O_3$ concentrations from the MetEmis experiment with the observed trends collected from Lin et al. (2016) and references therein. The observed trends of annual-mean $O_3$ at Beijing, Hongkong, Taiwan, Waliguan, and South Korea stations are +0.90 ppbv yr$^{-1}$, +0.58 ppbv yr$^{-1}$, +0.54 ppbv yr$^{-1}$, +0.25 ppbv yr$^{-1}$, +0.48 ppbv yr$^{-1}$, respectively. Simulated $O_3$ concentrations at all stations exhibit statistically significant increasing trends, although the model underestimates the trends. The modeled $O_3$ trends were reported to have low biases in previous studies (Tanimoto et al., 2009; Parrish et al., 2014). Parrish et al. (2014) compared $O_3$ trends simulated by three chemistry-climate models with observations at Asian sites, and reported that one model captured less than one third of the observed increasing trend whereas the other two models suggested no significant increasing trends."

[Figure]

**Figure 5.** Comparison of simulated trends in annual-mean $O_3$ concentrations from the MetEmis experiment with observations for Beijing (location: 40.0 °N, 116.5 °E; years: 1995–2005; reference: Ding et al., 2008a), Hongkong (22.2 °N, 114.3 °E; 1994–2007; Wang et al., 2009b), Taiwan (23.5 °N, 121.0 °E; 1994–2007; Lin et al., 2010), Waliguan (36.3 °N, 100.9 °E; 1994–2013; Xu et al. 2016), and South Korea (37.3 °N, 126.5 °E; 1990–2010; Lee et al., 2014).

(2) This study defines the Asian ozone outflow as the ozone flux through the meridional plain along 135E from 20N-55N and from the surface to 100 hPa. If you restrict the calculation to the surface to 200-300 hPa or up to the tropopause, does the calculated $O_3$ flux change substantially? I wonder if the $O_3$ flux up to 100 hPa is overwhelmingly influenced by stratosphere-to-troposphere exchange (STE) and thus the emission-driven trend is swamped by interannual variability in STE.

**Response:**

1) As we described in the last paragraph of Section 2.2, the interannual variations in stratosphere–troposphere exchange (STE) of $O_3$ is not considered in this work. The model imposes a global annual mean cross-tropopause $O_3$ flux of 500 Tg yr$^{-1}$. We have added the following discussions on this issue in the Uncertainty Discussion section (Section 6) of our revised manuscript: "There are some uncertainties in our simulations. First, the influence of interannual variation in stratosphere-troposphere exchange on tropospheric $O_3$ is not considered in this study. Terao et al. (2008) reported that the stratosphere-troposphere exchange had large impacts on the interannual variability of tropospheric $O_3$ over Canada and Europe but the impact was much smaller over East Asia."

2) Following the Reviewer's suggestion, we have calculated $O_3$ fluxes through the meridional plain along 135 °E from the surface to 200 hPa. Figures R1(a) and R1(b) show the evolutions of annual $O_3$ outflow fluxes across the meridional plane from the surface to 100 hPa and from the surface to

200 hPa, respectively. When we restrict the calculation to 200 hPa, the patterns of variations in $O_3$ fluxes are similar to those calculated from the surface to 100 hPa. Both figures show that, with variations in both anthropogenic emissions and meteorological parameters (the MetEmis simulation), the simulated $O_3$ outflow has large IAVs but a statistically insignificant ($P > 0.05$) trend.

[Figure]

**Figure R1.** Evolution of annual $O_3$ outflow fluxes (Tg yr$^{-1}$) across the meridional plane along 135 °E from 20 °N to 55 °N for (a) from the surface to 100 hPa and (b) from the surface to 200 hPa.

(3) This study uses tropospheric column ozone (TCO) retrieved from TOMS/SBUV to evaluate their model simulation of TCO seasonal cycle and long-term trends (Figs 3 and 4). But how good are the TOMS TCO retrievals? TOMS TCO is possibly representative of mid- and upper tropospheric ozone variability. It is not expected to resolve ozone variability in the lower troposphere. So why use TOMS to evaluate the model?

**Response:**

 As both reviewers pointed out, the TOMS/SBUV retrievals have large uncertainty in the lower troposphere $O_3$. Therefore, we have changed the comparisons with satellite to the comparisons with surface and sondes observations in the revised manuscript (the second and the third paragraphs of Section 3). The data from WDCGG and EANET are used to evaluate the simulated surface-layer $O_3$ concentrations, and the ozonesonde data from WOUDC are used to evaluate the simulated $O_3$ concentrations for the boundary layer, middle and upper troposphere. See our response to Major Comment (2) of Reviewer #2 for further details of comparisons.

(4) Fig.9 and associated discussions about the future changes. Changes in atmospheric circulation on regional scales under future climate scenarios are known to have large uncertainty. The different models often yield different results and large ensemble members are typically required.
Theodore G. Shepherd, Atmospheric circulation as a source of uncertainty in climate change projections, Nature Geoscience 7, 703–708 (2014), doi:10.1038/ngeo2253.
How many ensembles are included in your experiments? Rather than just showing your results, a through literature review is needed in Section 5 to place your results into context. What is the robust conclusion across the models in the published literature regarding changes in zonal winds and other circulation aspects over Asia under future climate? Do your model agree with the published work?

**Response:**
1) The future simulation of $O_3$ outflow is driven by the archived meteorological data from the Goddard Institute for Space Studies (GISS) general circulation model (GCM) 3. The GISS Model 3 is coupled with a "Q-flux" ocean as described in Wu et al. (2008). The GISS Model 3/GEOS-Chem combination has been used to project future concentrations of tropospheric ozone and aerosols over United States and China under future climate (Wu et al., 2008; Pye et al., 2009; Wang et al., 2013; Jiang et al., 2013).
2) Following the Reviewer's suggestion, we now compare future changes in zonal winds projected by our model with those reported in previous studies in Section 5.2 of our revised manuscript: "Our projected future changes in zonal winds are consistent with previous studies. By analyzing 18 CMIP5 models, Huang and Wang (2016) assessed the future changes in atmospheric circulation

during spring over East Asia. They found that, although different models projected different changes (even in sign) in zonal winds, the ensemble mean of five better-skill models among the 18 CMIP5 models exhibited overall increases in zonal winds throughout the whole troposphere during spring, which agrees with our simulation. Based on 31 (29)-model ensemble mean results, Jiang and Tian (2013) showed that the westerlies along 135 °E during winter (summer) were projected to weaken (strengthen). Such projected patterns of future changes in westerlies during winter and summer are also captured by our model." We have also added the following discussions on this issue in the Uncertainty Discussion section (Section 6) of our revised manuscript: "Finally, projecting future atmospheric circulation on regional scales has large uncertainty, which is undergoing continuing improvement."

**Other minor comments**

Page 2, Line 10: Also cite Jacob et al., 1999 - the first paper on Asian influence on US ozone.
Jacob, D. J., Logan, J. A. Murti, P. P. Effect of rising Asian emissions on surface ozone in the United States. Geophys. Res. Lett. 26, 2175-2178 (1999).
**Response:**
We have added the reference (Jacob et al., 1999) in the revised manuscript (the second paragraph of Section 1).

Page 3, Lines 9-10: Also cite Lin et al. (2015, GRL) - who found that measurement sampling biases substantially influence the ozone trends derived from sparse measurements over the western US originally reported by Cooper et al. (2010, Nature).
Lin, M., .W. Horowitz, O.R. Cooper, D. Tarasick, S. Conley, L.T. Iraci, B. Johnson, T. Leblanc, I. Petropavlovskikh, E.L. Yates (2015): Revisiting the evidence of increasing springtime ozone mixing ratios in the free troposphere over western North America, Geophysical Research Letter, 42, doi:10.1002/2015GL065311.
**Response:**
We have added the reference (Lin et al., 2015) in the revised manuscript (the fourth paragraph of Section 1).

Page 3, Lines 13-15: Also cite Schnell, J. L., M. J. Prather, B. Josse, V. Naik, L. W. Horowitz, G. Zeng, D. T. Shindell, and G. Faluvegi (2016), Effect of climate change on surface ozone over North America, Europe, and East Asia, Geophys. Res. Lett., 43, 3509-3518, doi: 10.1002/2016GL068060.
**Response:**
We have added the reference (Schnell et al., 2016) in the revised manuscript (the fourth paragraph of Section 1).

Page 3, Lines 20-30: Also explicitly discuss the results from Lin et al. (2014, Nature Geoscience). They found that interannual variability of springtime Asian ozone outflow is strongly influenced by ENSO-related shifts in the subtropical jet stream. Transport of Asian pollution towards the eastern North Pacific during spring has weakened in the 2000s due to more frequent La Nina-like conditions.
**Response:**
Following the Reviewer's suggestion, we have added the following discussions in the fourth paragraph of Section 1 in the revised manuscript: "Lin et al. (2014) reported that interannual variability in springtime Asian $O_3$ transport, as inferred by the East Asian COt (carbon-monoxide-like tracer), was strongly influenced by ENSO-related shifts in the subtropical jet stream, and that the decrease in ozone-rich Eurasian airflow reaching the eastern North Pacific during spring in the 2000s was attributed to more frequent La Nina events."

**Table 4.** Information for the sites with $O_3$ measurements used in model evaluation.

| Site | Location | Database | Height | R[a] | NMB[b] (%) |
|---|---|---|---|---|---|
| Minamitorishima | 24.3 N, 154.0 E | WDCGG | surface | 0.92 | +12.7 |
| Yonagunijima | 24.5 N, 123.0 E | WDCGG | surface | 0.93 | +12.6 |
| Rishiri | 45.1 N, 141.2 E | EANET | surface | 0.82 | +2.4 |
| Ogasawara | 27.1 N, 142.2 E | EANET | surface | 0.90 | +29.6 |
| Naha | 26.2 N, 127.7 E | WOUDC | 500–300 hPa | 0.68 | –2.61 |
| | | | 700–500 hPa | 0.77 | +16.4 |
| | | | 850–700 hPa | 0.85 | +24.3 |
| | | | 1000–850 hPa | 0.88 | +39.5 |
| Tsukuba | 36.1 N, 140.1 E | WOUDC | 500–300 hPa | 0.55 | +15.8 |
| | | | 700–500 hPa | 0.76 | +12.3 |
| | | | 850–700 hPa | 0.76 | +8.61 |
| | | | 1000–850 hPa | 0.60 | +8.5 |

[a] Correlation coefficient (R) between the observed and simulated monthly $O_3$ mixing ratios.

[b] Normalized mean bias (NMB, %) between the observed and simulated monthly $O_3$ mixing ratios.

[Figure]

**Figure 3.** Time series of monthly surface-layer O$_3$ mixing ratios measured by WDCGG and EANET (blue line), and simulated by MetEmisB (red line). (a) Minamitorishima and (b) Yonagunijima are WDCGG sites, and (c) Rishiri and (d) Ogasawara sites are EANET sites.

[Figure]

**Figure 4.** Time series of monthly $O_3$ mixing ratios measured by ozonesonde (blue line), and simulated by MetEmisB (red line). (a) Naha and (b) Tsukuba are ozonesonde sites from WOUDC. Comparisons are shown for four altitude levels in the troposphere.

(3) Is 135 degE appropriate? - The authors mentioned in the title "ozone outflow from East Asia" and used a longitudinal transect at 135 degE to diagnose the eastward flux of ozone. I wonder why at 135 degE, not 120 degE, to be more close to ozone production region in central-eastern China. I think, if the authors look at the flux at 120 degE, they would obtain higher signals in the ozone flux, and this would be much more direct in interpreting the model simulations. Also, the authors mainly discuss central-eastern China or North China Plain, rather than whole East Asia. This should be explicitly phrased, for example, "outflow from central-eastern China", since this paper is not looking at the impacts on the western North America but focusing on export region.

**Response:**

We calculate $O_3$ flux through the vertical plane along $135\,°\,E$, because $135\,°\,E$ is the easternmost boundary of China (i.e., Wusuli River in Northeastern China). Following the Reviewer's suggestion, we have also calculated the $O_3$ outflow flux along $120\,°\,E$. Figure H1 shows the evolutions of annual $O_3$

outflow fluxes across the meridional plane along (a) 135 °E and (b) 120 °E over 1986–2006 in the Met, Emis, and MetEmis simulations. The variations in $O_3$ fluxes calculated at 120 °E are similar to those calculated at 135 °E. Both figures show that, with variations in both anthropogenic emissions and meteorological parameters (the MetEmis simulation), the simulated $O_3$ outflow shows large IAVs but a statistically insignificant ($P > 0.05$) trend. We have added the above discussion in the Uncertainty Discussion section (Section 6) of our revised manuscript.

Because the variations in $O_3$ fluxes calculated at 135 °E are similar to those calculated at 120 °E, we retain the calculations along 135 °E and the description of "ozone outflow from East Asia" in the revised manuscript.

[Figure]

**Figure H1.** Evolution of annual $O_3$ outflow fluxes (Tg yr$^{-1}$) across the meridional plane along (a) 135 ° E, and (b) 120 °E, from 20 °N to 55 °N and from the surface to 100 hPa over 1986–2006 in the Met, Emis, and MetEmis simulations.

**Specific comments**:

Abstract, L22: insignificant decadal trend of -2.2%/decade. Add +/- uncertainty, or just delete the number here.
**Response:**
    We have deleted the number in our revised manuscript.

L28-29: spring and summer. The maritime flow from the Pacific Ocean is predominant in summer. Is summer really effective in the enhancement of continental outflow? I do not see strong enhancement in summer in Figure 9.
**Response:**
    Although the absolute value of the increase in $O_3$ outflow during summer is not large, the percentage increase in $O_3$ outflow during summer is 14.5% (Table 5 in the revised manuscript). The large percentage increase can be mainly attributed to the enhancement in zonal winds. Based on 29-model ensemble mean results, Jiang and Tian (2013) also showed that the westerlies along 135 °E during summer would strengthen in future climate.

L31: important implications for long-term air quality planning. For whom? For US? For northern midlatitudes? For China? For East Asia?
**Response:**
    We have revised the sentence as "have important implications for long-term air quality planning for the downwind regions of China, such as Japan and US" in the Abstract of our revised manuscript.

P2, L8-11: . . . influences ozone air quality in the downwind regions, such as the US and Canada. Downwind regions are not only the western US, but should include the neighboring regions and the Pacific Ocean. Ou-Yang et al. paper is already cited here, so the sentence should be rephrased to be something like ". . . such as the western North Pacific through the western North America", and add some other references, for example, papers reporting long-range transport to Korea, Japan, and the

Northern Pacific (Han et al., ACP, 2015; Tanimoto et al., GRL, 2005; Pochanart et al., 2015, and many others!).

**Response:**

We have replaced it by "such as the western North Pacific through the western North America", and added the following references: "Tanimoto et al., 2005; Kim et al., 2006; Li et al., 2008; Kurokawa et al., 2009; Nagashima et al., 2010; Han et al., 2015; Pochanart et al., 2015" (the second paragraph of Section 1).

P3, L8-12: Tanimoto, AE, 2009 should be cited here (decadal trends of . . .)

**Response:**

We have cited the reference (Tanimoto, 2009) in the revised manuscript (the fourth paragraph of Section 1).

Figure 4: The authors showed comparison of TCO for GEOS-Chem and TOMS/SBUV, suggesting the biases in the model. Again, why don't the authors make comparison to the surface and sonde observations?

**Response:**

As suggested by the Reviewer, we have conducted comparisons with surface-layer $O_3$ measurements from WDCGG and EANET, and with ozonesonde measurements from WOUDC for the boundary layer, middle and upper troposphere $O_3$ in the revised manuscript. See our response to your major comment (2).

Also, in P8, L10-13, the authors state that "although GEOS-Chem overestimates TCO values over eastern China and the western Pacific Ocean, the model exhibits reasonable performance in simulating the spatiotemporal distributions of the tropospheric ozone column burden over China and downwind regions, which lends us confidence to simulate the temporal evolutions of the Asian ozone outflow." I would not agree with this statement, since model and satellite are quite different in the tail of outflow from China (the region of >40 DU, shown in orange), and this difference would lead to large biases in calculating outflow flux in particular, as the authors set the diagnosis line at 135 degE, off China, and over Japan. Also, technically, the authors said "the western Pacific Ocean" here and also in the Figure 4 caption, but the region where the authors pointed is mostly Japan, so the description must be accurately modified.

**Response:**

As the Reviewer pointed out in major comment (2), because of the large uncertainty in the retrieval of tropospheric ozone, comparison to satellite is not a robust way to quantitatively evaluate the model performance. Therefore, we have changed the comparisons with satellite to the comparisons with surface and sondes observations in the revised manuscript. See our response to your major comment (2). The old Figure 4 and associated description have been deleted and replaced by the comparisons with surface and sondes observations.

P9, Section 4.2 IAV and decadal trends: The authors basically said that the influence of Met. is larger than Emiss., which makes sense if they diagnosed at 135 degE, off the Asian continent, where Asian monsoon impacts are substantial.

**Response:**

Following your suggestions in major comment (3), we have calculated the $O_3$ outflow flux along $120\,°E$, more close to the Asian continent. It is concluded from Figure H1 that the variations in $O_3$ fluxes calculated at $120\,°E$ are similar to those calculated at $135\,°E$. With variations in both anthropogenic emissions and meteorological parameters (the MetEmis simulation), the simulated $O_3$ outflow along $120\,°E$ also shows large IAVs but a statistically insignificant ($P > 0.05$) trend. The two curves from the Met and MetEmis simulations almost coincide with each other, indicating the dominant role of variations in meteorological parameters in the IAVs of the Asian $O_3$ outflow flux.

A number of important references are missing:
Pochanart, P. et al, 2015, Boundary Layer Ozone Transport from Eastern China to Southern Japan: Pollution Episodes Observed during Monsoon Onset in 2004, Asian J. Atmos. Environ. 9, 48-56.

Tanimoto, H., et al., 2005, Significant latitudinal gradient in the surface ozone spring maximum over East Asia, Geophys. Res. Lett. 32, L21805.

Tanimoto, H., 2009, Increase in springtime tropospheric ozone at a mountainous site in Japan for the period 1998–2006, Atmos. Environ., 43, 1358-1363.

Nagashima, T., et al., 2010, The relative importance of various source regions on East Asian surface ozone, Atmos. Chem. Phys., 10, 11305-11322, doi:10.5194/acp-10-11305-20109077-9120.

Li, J., et al., 2008, Regional-scale modeling of near-ground ozone in the Central East China, source attributions and an assessment of outflow to East Asia - The role of regional-scale transport during MTX2006, Atmos. Chem. Phys., 8, doi: 10.5194/acpd-8-13159-2008.

Tanimoto, H., et al., 2015, Consistency of tropospheric ozone observations made by different platforms and techniques in the global databases, Tellus B, 67, 27073, doi:10.3402/tellusb.v67.27073.

**Response:**

    We have added the above references in proper places of the revised manuscript.

[revised manuscript text omitted]

---

## Author Response (AR2)

**Responses to reviewer's comments on "Interannual variation, decadal trend, and future change in ozone outflow from East Asia" by J. Zhu et al. (MS No.: acp-2016-938)**

We would like to thank Reviewer #1 for the additional comments that are helpful for improving the quality of our work. The manuscript has been revised accordingly, and our point-by-point responses are provided below. The reviewer's comments are shown in black and our replies including the updates to the manuscript are highlighted in blue below. A marked-up manuscript version is also showing the changes made.

**Response to Reviewer #1**

1. Abstract, around L20: Before discussing the model attribution, please add a few sentences stating how well the model captures the observed ozone interannual variability (r-squared = ?) and that the model significantly underestimates long-term increases in surface ozone over East Asia. Such statements should also be included in the Conclusion before discussing any attribution results.
**Response:**
Following the Reviewer's suggestion, we have added the following description in the Abstract: "Evaluation of the model results against measurements shows that the GEOS-Chem model captures fairly well the seasonal cycles and interannual variations of tropospheric $O_3$ concentrations, with high correlation coefficients of 0.82–0.93 at four ground-based sites and of 0.55–0.88 at two ozonesonde sites where observations are available. The increasing trends in surface-layer $O_3$ concentrations in East Asia over the past two decades are captured by the model, although the modeled $O_3$ trends have low biases." Similar statements have also been added in the Conclusion.

2. Introduction, Page 3, around Line 19: Move the discussions of future ozone changes to the next paragraph and add a few sentences at the end of the previous paragraph to clearly state (1) Asian NOx emissions have more than tripled over the past 20-30 years (Granier et al., 2011; Zhang Q. et al., HillBoll et al., 2013); (2) Consequently, outflow of Asian pollution has increased significantly and has contributed to raising springtime ozone observed over western North America (Refs, e.g., Lin et al., act-2016-1093).
In the response to reviewers, the authors stated that there is no significant trend in Asian ozone outflow because the offsetting effects of North American and European emission reductions. However, you are looking at outflow of ozone immediately downwind of Asia, which should be dominated by the trends of Asian NOx emissions. The early modeling results of Lamarque et al. (2010) and Parrish et al. (2014) are based on free-running chemistry-climate models that generate their own meteorology and thus are not expected to reproduce the influence of meteorological variability on observed ozone trends. As the authors nicely demonstrated in this manuscript, the meteorologically-driven ozone interannual variability is quite large. Therefore, the trend analysis presented in this article (e.g., Fig.5) should be compared to more recent work using long-term model simulations driven by observed meteorology.
**Response:**
We have moved the discussions on future $O_3$ changes to the next paragraph in the revised manuscript. Over East-Central China (the most polluted region), $NO_x$ emissions have tripled over the past 20 years (Granier et al., 2011; Hilboll et al., 2013); while Asian $NO_x$ emissions almost doubled over the past 20 years (Yang et al., 2015). Therefore, we have added the following sentences in the revised manuscript: "Asian $NO_x$ emissions almost doubled over the past 20 years (Yang et al., 2015), which contributed to the raised $O_3$ observed over the downwind regions of Asia (Lin et al., 2016)."

Tropospheric $O_3$ has relatively long lifetime (~3 weeks) (Fiore et al., 2002; Liao et al., 2006), therefore Asian $O_3$ outflow indeed includes the effects of emissions in different regions of the world, other than Asian emissions alone. Wang et al. (2011) reported that annual mean surface $O_3$ averaged over China from anthropogenic emissions outside of China and from Chinese anthropogenic emissions were 12.6 ppbv and 5.4 ppbv respectively, suggesting the important role of anthropogenic emissions outside of China in $O_3$ levels over China.

A recent study by Strode et al. (2015) investigated the observed trends in surface $O_3$ using a chemical transport model Global Modeling Initiative (GMI) driven by the GMAO reanalysis

meteorology MERRA, and found that the model also underestimated the observed positive trends for 1991–2010. We have added this reference in the revised manuscript.

3. Specific comments on the figures and associated discussions:

Figure 1: please discuss how the trends in $NO_x$ emissions used in this study compare to the trends derived from satellite $NO_2$ columns for the overlapping period 1996-2005? Can the $NO_x$ emission biases explain the biases in simulated ozone trends shown in Figure 5?

**Response:**

   Following the Reviewer's suggestion, we have added the following sentences at the end of the second paragraph of Section 2.2: "Note that, over 1996–2006 when $NO_x$ emissions and satellite $NO_2$ columns were simultaneously available, the trend in $NO_x$ emissions over East-Central China (ECC, 110–123 °E, 30–40 °N) was +8.2% $yr^{-1}$ on the basis of the emission inventory used in this study, close to the trend of +9.0% $yr^{-1}$ in $NO_2$ columns averaged over ECC on the basis of tropospheric $NO_2$ vertical column density (VCD) data retrieved from GOME (1996–2002) and SCIAMACHY (2003–2006), which are available from www.temis.nl."

   We have also compared the Asian (60–150 °E, 10 °S–55 °N) $NO_x$ emissions in this study with other emissions inventories that have multiple years of emissions available; the decadal trends in this work (years 1986–2006), REAS v1.1 (years 1986–2003), and EDGAR v4.2 (years 1986–2006) are similar, with values of +2.4, +2.8, and +2.7 Tg N $decade^{-1}$, respectively. The slight underestimation of $NO_x$ emissions in our work should not compromise our model results.

Figure 3 and Table 4: The correlations reported in Table 4 mainly reflect the prominent variability on monthly/seasonal time scales, as opposed to interannual time scales. It is important to evaluate how well the model captures the observed interannual variability – the focus of the paper. Thus, the referee suggests conducting the comparison and interannual correlations for each season (DJF, MAM, JJA, SON) over the study period.

**Response:**

   The correlations reported in this study by comparing monthly $O_3$ data over 10 years may reflect the variations on both seasonal and interannual time scales. We do not conduct comparison by calculating interannual correlations for each season because the sample size for comparison in this way is small (i.e., 10 pairs for each season and each site), while comparing monthly $O_3$ data over 10 years provides enough samples (i.e., 120 pairs for each site) so that we can conduct a robust comparison. Therefore, we retain the correlations reported in this study, which may reflect both seasonal cycles and interannual variations.

Figure 4: The ozonesondes sampling frequency is less than 4 to 5 profiles per month at these sites. These weekly ozonesondes are too infrequent to capture the actual interannual variability and long-term trends of seasonal mean ozone in surface air and aloft, given the large meteorologically-driven ozone variability. The referee suggests only using ozonesonde data for evaluating the simulated monthly mean ozone climatology.

**Response:**

   The model-observation comparison for the surface and aloft $O_3$ concentrations by using surface and ozonesondes measurements available in East Asia was suggested by Reviewer #2. The ozonesonde data could provide $O_3$ concentrations for the boundary layer, middle, and upper troposphere, and therefore the data were widely used by previous studies (Zhang et al., 2008; Walker et al., 2010). Tanimoto et al. (2015) evaluated the consistency of tropospheric $O_3$ observations made by means of multiple platforms, and found good agreement between ozonesonde and surface observations for the lower troposphere $O_3$. Therefore, we retain the comparison with ozonesonde.

Figure 5: Again, the comparison of observed and simulated ozone trends should be made on a seasonal basis!

**Response:**

   Following the Reviewer's suggestion, we now compare observed and simulated trends on a

seasonal basis for sites with observed seasonal trends available in Section 3 of our revised manuscript: "Figure 5 compares the simulated trends in seasonal or annual mean surface-layer $O_3$ concentrations from the MetEmis experiment with the observed trends collected from previous studies. Simulated $O_3$ concentrations exhibit statistically significant increasing trends at all sites except for Waliguan in winter, although the model underestimates the trends for some stations and seasons. The modeled $O_3$ trends were also reported to have low biases in previous studies (Tanimoto et al., 2009; Parrish et al., 2014; Strode et al., 2015)."

[Figure]

**Figure 5**. Comparison of simulated trends in seasonal or annual mean surface-layer $O_3$ concentrations from the MetEmis experiment with observations for Hongkong (location: 22.2 °N, 114.3 °E; years: 1994–2007; reference: Wang et al., 2009b), Waliguan (36.3 °N, 100.9 °E; 1994–2013; Xu et al. 2016), Beijing (40.0 °N, 116.5 °E; 2001–2006; Tang et al., 2009), and Taiwan (23.5 °N, 121.0 °E; 1994–2007; Lin et al., 2010). The simulated trend at Waliguan site for winter is statistically insignificant. The trends in seasonal-mean $O_3$ concentrations at Taiwan station are unavailable.

Figures 6 and 7: It would be very useful to the readers of the paper if the authors could show a pressure-latitude cross-section similar to Figure 6, but for the linear trends of seasonal mean ozone in ppb yr$^{-1}$ over 1986-2006 (highlight where the trends are statistically significant at the 95% confidence level using boxes or stippling). You can then discuss the vertical and latitudinal distribution of changes in Asian pollution outflow during the historical period. The proposed analysis will also provide additional insights into whether the lack of significant ozone trends currently shown in Figure 7 is due to spatial averaging.
**Response:**
    It is noted that this paper is focused on the variations in $O_3$ outflow fluxes, rather than $O_3$ concentrations. Therefore, we show in Figure R1 the pressure-latitude cross-sections of the linear trends in seasonal $O_3$ outflow fluxes from the MetEmis experiment over 1986–2006 (highlight where the trends are statistically significant at the 95% confidence level using stippling). The seasonal $O_3$ outflow fluxes do not show statistically significant trends almost everywhere (Figure R1), which agrees with the conclusion drawn from Figure 7. Therefore, the lack of significant trends in $O_3$ outflow fluxes shown in Figure 7 is not owing to spatial or temporal averaging. The Asian $O_3$ outflow show indeed statistically insignificant decadal trend, with variations in both anthropogenic emissions and meteorological parameters.
    Because the seasonal $O_3$ outflow fluxes do not show statistically significant trends almost everywhere and the conclusion drawn from Figure R1 agrees with that drawn from Figure 7, we do not add Figure R1 in the revised manuscript.

[Figure]

**Figure R1**. The pressure–latitude cross-sections along $135\,^\circ$ E of the linear trends in seasonal $O_3$ outflow fluxes from the MetEmis experiment over 1986–2006 (units: kg season$^{-1}$ m$^{-2}$). The dotted areas are statistically significant at the 95 % confidence level.

4. Discussion on the influence of climate change: In fact, many studies suggest that a warming climate would most likely worsen regional air stagnation events and thus decrease outflow of pollution from a source region. Your conclusion that future climate change will lead to increases in Asian ozone outflow in spring and summer seems to contradict with the findings in the published literature. Furthermore, increasing water vapor in a warming climate will lead to lower ozone at remote locations.
**Response:**
    Stagnation occurrence is projected to increase over many tropical and subtropical regions; but robust changes do not emerge over China in the middle 21st century (Horton et al., 2014). The projected increases in $O_3$ outflow fluxes during spring and summer mainly result from the increases in zonal winds during the two seasons. Our projected future increases in zonal winds are consistent with previous studies (Jiang and Tian, 2013; Huang and Wang, 2016). Huang and Wang (2016) assessed the future changes in atmospheric circulation during spring over East Asia, and found that the ensemble mean of five better-skill models among the 18 CMIP5 models exhibited overall increases in zonal winds throughout the whole troposphere during spring. Based on 29-model ensemble mean results, Jiang and Tian (2013) showed that the westerlies along 135 °E during summer were projected to strengthen.
    A warming climate is predicted to reduce tropospheric $O_3$ in remote regions by higher temperature and water vapor, but to increase $O_3$ over populated areas as a result of enhanced biogenic hydrocarbon emissions, and decomposition of peroxyacetyl nitrate at higher temperature (Katragkou et al., 2011; Rasmussen et al., 2012; Doherty et al., 2013; Kim et al., 2014). This paper focuses on $O_3$ outflow from East Asia, which is a highly populated region.

[revised manuscript text omitted]

---

## Author Response (AR3)

**Response to Co-Editor**

**Co-Editor Decision: Publish subject to technical corrections** (02 Mar 2017) by Yugo Kanaya

Comments to the Author:

Dear Authors,

Thank you for your revision. I find the authors made adequate revisions in response to the reviewer's comments.

One technical suggestion:

The sentences newly inserted in Abstract (from line20 to 24) should be better located before the sentence "Sensitivity studies are conducted to ..." starting from line 19.

I would appreciate if the authors could take this into account.

Best regards,

Yugo Kanaya, ACP Co-editor

**Response:**

We would like to thank Co-Editor for the suggestion. We now have located the newly inserted sentences before the sentence "Sensitivity studies are conducted to ..." in the Abstract.

[revised manuscript text omitted]